# A DIFFERENTIABLE RANK-BASED OBJECTIVE FOR BETTER FEATURE LEARNING

**Krunoslav Lehman Pavasovic**[*]
Meta FAIR, Paris

**Giulio Biroli**[†]
ENS Paris

**Levent Sagun**[†]
Meta FAIR, Paris

## ABSTRACT

In this paper, we leverage existing statistical methods to better understand feature learning from data. We tackle this by modifying the model-free variable selection method, Feature Ordering by Conditional Independence (FOCI), which is introduced in Azadkia & Chatterjee (2021). While FOCI is based on a non-parametric coefficient of conditional dependence, we introduce its parametric, differentiable approximation. With this approximate coefficient of correlation, we present a new algorithm called difFOCI, which is applicable to a wider range of machine learning problems thanks to its differentiable nature and learnable parameters. We present difFOCI in three contexts: (1) as a variable selection method with baseline comparisons to FOCI, (2) as a trainable model parametrized with a neural network, and (3) as a generic, widely applicable neural network regularizer, one that improves feature learning with better management of spurious correlations. We evaluate difFOCI on increasingly complex problems ranging from basic variable selection in toy examples to saliency map comparisons in convolutional networks. We then show how difFOCI can be incorporated in the context of fairness to facilitate classifications without relying on sensitive data.

## 1 INTRODUCTION

Feature learning is crucial in machine learning (ML), enabling models to learn meaningful representations of the data. It can improve performance, reduce dimensionality, increase interpretability, and provide flexibility for adapting to new data distributions and tasks (Bengio et al., 2012; 2013). However, increasing model transparency (Arrieta et al., 2020; Räuker et al., 2023), improving disentanglement and understanding architectural biases (Bouchacourt et al., 2021; Roth et al., 2022), as well as learning invariances to improve robustness (Arjovsky et al., 2019) have proven to be challenging.

In this paper, we propose a new feature-learning approach that relies on ranks, notion seldom explored in the literature but thoroughly studied in statistics. The importance of ranks is evident, from independence tests (Bergsma & Dassios, 2014; Blum et al., 1961; Csörgő, 1985; Deb & Sen, 2023; Drton et al., 2020) and sensitivity analysis (Gamboa et al., 2018), to multivariate analysis (Sen & Puri, 1971) and measuring deviation (Rosenblatt, 1975). However, most of these methods are nonparametric and, therefore, not easily extendable to feature learning with neural networks (NNs). While there are a handful of feature learning works that rely on rank notions (Kuo & Hsu, 2017; Wojtas & Chen, 2020; Fan et al., 2023; Li et al., 2023b), these works do so indirectly and through reliance on two NNs; one that optimizes for a non-rank-based feature-learning objective and another that learns how to rank those learned features according to some similarity measure.

To fill this gap, we propose difFOCI, a parametric relaxation of the nonparametric, rank-based measures of correlation (Chatterjee, 2020; Azadkia & Chatterjee, 2021), which generalizes the measure proposed by Dette et al. (2013), roots of the idea can be related to the Rényi correlation (Bickel et al., 1993; Rényi, 1959). To the best of our knowledge, difFOCI is the first parametric framework that directly optimizes a rank-based objective, making it directly applicable to numerous applications in machine learning, including end-to-end trainable neural networks. We demonstrate that difFOCI yields strong results in various areas, including *(i)* feature selection, *(ii)* domain shift and spurious correlation, and *(iii)* fairness experiments.

---

[*]Correspondence to krunolp@meta.com. [†] Joint last author.

**Organization of the paper.** In Section 2, we introduce the notation and technical background. In Section 3, we outline the main results of this paper, explaining the proposed metric, and establishing its theoretical properties. We analyze it in toy examples that demonstrate solid performance. In Section 4, we extend difFOCI, showcasing its strengths in three examples. In Section 5, we highlight its wide applicability to real-world data, showing it achieves state-of-the-art performance on feature selection and dimensionality reduction, and competitive performance in domain shift and fairness literature. Finally, in Section 6, we conclude with a few remarks on the potential future applications.

## 2 PRELIMINARIES AND TECHNICAL BACKGROUND

### 2.1 NOTATION AND PRELIMINARY DEFINITIONS

We let $I_d$ denote the $d \times d$ identity matrix and $[n] = \{1, \ldots, n\}$. We let $S(A) = \pi_1(A), \ldots, \pi_{n!}(A)$ be the set of all permutations of a set $A$, with $|A| = n$. For a matrix $\mathbf{X}$, we denote the set of all permutations its columns by $S(\mathbf{X})$ and by $\pi_i^j(\mathbf{X})$, we represent the $i$-th element of the $j$-th permutation. We denote its $p$-th through $q$-th column as $\mathbf{X}_{p:q}$, with $p > q$, $p, q \in \mathbb{N}$. We define the Hadamard product between a vector $\alpha \in \mathbb{R}^p$ and a matrix $\mathbf{X} \in \mathbb{R}^{n,p}$ as $(\alpha \odot \mathbf{X})_{i,j} := \alpha_i \mathbf{X}_{i,j}$. We represent the scaled Softmax function with $\sigma_\beta(x)$, where $\sigma_\beta(x)_i = e^{\beta x_i} / \sum_{j=1}^d e^{\beta x_j}$, for $x \in \mathbb{R}^d$, $\beta \in \mathbb{R}^+$. Finally, we use $c(x, p)$ to denote zeroing out any $x_i$ with $|x_i| \leq p$, for $x \in \mathbb{R}^d$ and $p \in \mathbb{R}$.

### 2.2 CHATERJEE'S COEFFICIENT

We present the novel rank-based estimator developed by Chatterjee (2020), which is the first of two foundational works necessary for our approach. Consider a random vector $(X, Y)$ on a probability space $(\Omega, \mathcal{F}, \mathbb{P})$, with $Y$ being non-constant and governed by the law $\mu$. The estimator approximates the following rank-based measure (Dette et al., 2013):

$$\xi(X, Y) := \frac{\int \text{Var} \left( \mathbb{E} \left( \mathbb{1}_{\{Y \geq t\}} \mid X \right) \right) d\mu(t)}{\int \text{Var} \left( \mathbb{1}_{\{Y \geq t\}} \right) d\mu(t)}. \tag{1}$$

Chatterjee (2020) establishes a straightforward estimator for (1) that has simple asymptotic theory, enjoys several consistency results and exhibits several natural properties; *(i)* normalization: $\xi(X, Y) \in [0, 1]$, *(ii)* independence: $\xi(X, Y) = 0 \iff Y \perp\!\!\!\perp X$, *(iii)* complete dependence: $\xi(X, Y) = 1 \iff Y$ a measurable function of $X$ a.s., and *(iv)* scale invariance: $\xi(aX, Y) = \xi(X, Y), a \in \mathbb{R}^*$. To estimate $\xi$, consider i.i.d. pairs $(X_i, Y_i)_{i=1}^n \sim (X, Y)$, with $n \geq 2$. Rearrange the data as $(X_{(1)}, Y_{(1)}), \ldots, (X_{(n)}, Y_{(n)})$, such that $X_{(1)} \leq \cdots \leq X_{(n)}$, breaking ties uniformly at random. Define $r_i$ as the rank of $Y_{(i)}$, i.e., the number of $j$ for which $Y_{(j)} \leq Y_{(i)}$, and $l_i$ as the number of $j$ such that $Y_{(j)} \geq Y_{(i)}$. The estimator is then defined as:

$$\xi_n(X, Y) := 1 - \frac{n \sum_{i=1}^{n-1} |r_{i+1} - r_i|}{2 \sum_{i=1}^n l_i (n - l_i)}. \tag{2}$$

Furthermore, Chatterjee (2020) establishes the following consistency result for $\xi_n$:

**Theorem 1.** *(Chatterjee, 2020) If $Y$ is not almost surely a constant, then as $n \to \infty$, $\xi_n(X, Y)$ converges almost surely to the deterministic limit $\xi(X, Y)$.*

In simulations by Chatterjee (2020), this estimator demonstrates greater efficacy than most signal-detection tests. Its applications span diverse areas: approximate unlearning (Mehta et al., 2022), topology (Deb et al., 2020), black carbon concentration estimation (Tang et al., 2023), sensitivity analysis (Gamboa et al., 2022), and causal discovery (Li et al., 2023a). Extensive further research has been conducted: its limiting variance under independence (Han & Huang, 2022), permutation testing (Kim et al., 2022), bootstrapping (Lin & Han, 2024), rate efficiency (Lin & Han, 2023), minimax optimality (Auddy et al., 2023) and kernel extension (Huang et al., 2022); Bickel (2022) analyzed it for independence testing, showing it might have no power or prove misleading[1].

---

[1]This does not impact us, however, as we do not utilize it for independence testing.

## 2.3 EXTENDING THE COEFFICIENT FOR ESTIMATING CONDITIONAL DEPENDENCE

In a subsequent study, Azadkia & Chatterjee (2021) extend the coefficient (1) $\xi$ to a measure $T(Y, \mathbf{Z} \mid \mathbf{X})$, capturing the strength of the conditional dependence between $Y$ and $\mathbf{Z}$, given $\mathbf{X}$. $T$ can be interpreted as a non-linear extension of the partial $R^2$ statistic (Draper & Smith, 1998), and reads as follows:

$$T = T(Y, \mathbf{Z} \mid \mathbf{X}) := \frac{\int \mathbb{E}(\mathrm{Var}(\mathbb{P}(Y \geq t \mid \mathbf{Z}, \mathbf{X}) \mid \mathbf{X})) d\mu(t)}{\int \mathbb{E}\left(\mathrm{Var}\left(\mathbb{1}_{\{Y \geq t\}} \mid \mathbf{X}\right)\right) d\mu(t)},$$

where $Y$ denotes a random variable governed by $\mu$, and $\mathbf{X} = (X_1, \ldots, X_p)$ and $\mathbf{Z} = (Z_1, \ldots, Z_q)$ are random vectors, defined within the same probability space, with i.i.d. copies $(\mathbf{X}_i, \mathbf{Z}_i, Y_i)_{i=1}^n \sim (\mathbf{X}, \mathbf{Z}, Y), n \geq 2$. Here, $q \geq 1$ and $p \geq 0$, with $p = 0$ indicating $\mathbf{X}$ has no components.

The statistic $T$ generalizes the univariate measure in (1). To construct its estimator, for each index $i$, define $N(i)$ as the index $j$ where $\mathbf{X}_j$ is the closest to $\mathbf{X}_i$, and $M(i)$ as the index $j$ where the pair $(\mathbf{X}_j, \mathbf{Z}_j)$ is closest to $(\mathbf{X}_i, \mathbf{Z}_i)$ in $\mathbb{R}^{p+q}$ w.r.t. the Euclidean metric and resolving ties randomly. The estimate of $T$ is given by:

$$T_n = T_n(Y, \mathbf{Z} \mid \mathbf{X}) := \frac{\sum_{i=1}^n \left(\min\left\{r_i, r_{M(i)}\right\} - \min\left\{r_i, r_{N(i)}\right\}\right)}{\sum_{i=1}^n \left(r_i - \min\left\{r_i, r_{N(i)}\right\}\right)}. \tag{3}$$

with $M(i)$ denoting the index $j$ such that $\mathbf{Z}_j$ is the nearest neighbor of $\mathbf{Z}_i$, $p \geq 1$ and $r_i, l_i$ as defined in Sec. 2.2[2]. The authors establish the same four natural properties for $T$ as for the estimator in (1) - normalization, independence, complete dependence, and scale invariance:

**Theorem 2.** *(Azadkia & Chatterjee, 2021) Suppose that $Y$ is not almost surely equal to a measurable function of $\mathbf{X}$. Then $T$ is well-defined and $0 \leq T \leq 1$. Moreover, $T = 0$ iff $Y$ and $\mathbf{Z}$ are conditionally independent given $\mathbf{X}$, and $T = 1$ iff $Y$ is almost surely equal to a measurable function of $\mathbf{Z}$ given $\mathbf{X}$.*

The authors further demonstrate that $T_n$ is indeed a consistent estimator of $T$:

**Theorem 3.** *(Azadkia & Chatterjee, 2021) Suppose that $Y$ is not almost surely equal to a measurable function of $\mathbf{X}$. Then as $n \to \infty, T_n \to T$ almost surely.*

## 2.4 FOCI: A NEW PARADIGM FOR FEATURE SELECTION

Azadkia & Chatterjee (2021) utilize the estimator $T_n$ to propose a novel, model-independent, step-wise feature selection method. The method, termed FOCI: Feature Ordering by Conditional Independence, is free from tuning parameters and demonstrates provable consistency. FOCI is outlined in Alg. 1, where we observe its iterative nature: variables are chosen one by one until the estimator's value drops below zero.

---

**Algorithm 1** FOCI

---

**Input:** $n$ i.i.d. copies of $(Y, \mathbf{X})$, with the set of predictors $\mathbf{X} = (X_j)_{j \in [p]}$ and response $Y$
$j_1 \leftarrow \arg\max_{j \in [p]} T_n(Y, X_j)$
**if** $T_n(Y, X_{j_1}) \leq 0$ **then**
    $\hat{S} = \emptyset$
**else**
    **while** $T_n\left(Y, X_j \mid X_{j_1}, \ldots, X_{j_k}\right) > 0$ **do**
        $j_{k+1} \leftarrow \arg\max_{[p] \setminus \{j_1, \ldots, j_k\}} T_n\left(Y, X_j \mid X_{j_1}, \ldots, X_{j_k}\right)$
    $\hat{S} = \{j_1, \ldots, j_{k'}\}$
**Output:** Set $\hat{S}$ of chosen predictors' indices

---

[2]The expression for $p = 0$ is given in Appendix A.1.

FOCI performs well on both simulated and real-world datasets. In a toy example with $Y = X_1 X_2 + \sin(X_1 X_3)$, where $X_i \sim \mathrm{N}(0, \sigma^2 \, \mathrm{I}_p)$, $\sigma^2 = 1$, and $i \in [2000], p = 100$, FOCI selects the correct subset 70 percent of the time. In contrast, popular scikit-learn feature selection algorithms (Pedregosa et al., 2011), explained in Sec. 5, almost never identify the correct subset (difFOCI, proposed in the next section, consistently selects the correct subset while preserving the same relative feature importance as FOCI during its correct runs). When applied to real-world datasets, FOCI matches the performance of established methods while requiring up to four times fewer features.

## 2.5 EXTENDING $T$ TO MACHINE AND DEEP LEARNING

From a statistical point of view, both $\xi_n$ and $T_n$ exhibit several strengths: well-established theoretical properties, are non-parametric, have no tunable parameters nor any distributional assumptions. Furthermore, a simple application of $T_n$ results in a strong feature-selection baseline. However, the non-smooth nature of the objectives in (2) and (3) renders them non-differentiable, and therefore not applicable to most ML applications[3].

In the following section, we make these objectives differentiable using straightforward, well-known tricks in the ML community. This allows us to extend them to various ML and deep learning applications (as showcased in Sec. 5). Moreover, it also allows to account for interactions between all features simultaneously (rather than in a step-wise fashion as in FOCI). Although FOCI could account for this in principle, as can be seen from Alg. 1, this would increase FOCI's complexity from $O(p^2)$ to potentially $O(2^p)$ thus preventing its practical use.

## 3 MAIN RESULTS

We now propose an alternative formulation to the estimator $T_n$ in (3), the objective of FOCI. As we will show later, this variation allows for the retention of FOCI's strengths as well as the improvement of its shortcomings.

## 3.1 DIFFOCI: TOWARDS A DIFFERENTIABLE VERSION OF FOCI

The initial step involves making the objective $T_n(Y, \mathbf{Z}|\mathbf{X})$ differentiable w.r.t inputs $\mathbf{Z}$. Implementing this can be accomplished using straightforward techniques. We employ the following approach:

1. Compute the pairwise distance matrix $\mathbf{M} \in \mathbb{R}^{n,n}$ where $M_{i,j} = \|\mathbf{X}_i - \mathbf{X}_j\|$.
2. Calculate $\mathbf{S}_\beta \in \mathbb{R}^{n,n}$ such that $\mathbf{S}_\beta = \sigma_\beta(-(\mathbf{M} + \lambda \, \mathrm{I}_n))^4$.
3. Instead of indexing $r_{N(i)} = r[N(i)]$, utilize $r^\top \mathbf{S}_{\beta_{i,\cdot}}$.

Similarly, for $\mathbf{U}_\beta := \sigma_\beta(-(\hat{\mathbf{M}} + \lambda \, \mathrm{I_n}))$, and $\hat{M}_{i,j} = \|(\mathbf{X}_i, \mathbf{Z}_i) - (\mathbf{X}_j, \mathbf{Z}_j)\|$. This allows us to present difFOCI, a differentiable version of the estimator in (3):

$$T_{n,\beta} = T_{n,\beta}(Y, \mathbf{Z}|\mathbf{X}) := \frac{\sum_{i=1}^n (\min\{r_i, r^\top \mathbf{U}_{\beta_{i,\cdot}}\} - \min\{r_i, r^\top \mathbf{S}_{\beta_{i,\cdot}}\})}{\sum_{i=1}^n (r_i - \min\{r_i, r^\top \mathbf{S}_{\beta_{i,\cdot}}\})}. \tag{4}$$

Using the following theorem, we establish that our new estimator (4) enjoys the same limiting theoretical properties as the estimator in (3):

**Theorem 4.** *Let $\beta \in \mathbb{R}^+$. Suppose that $Y$ is not almost surely equal to a measurable function of $\mathbf{X}$. Then, $\lim_{n\to\infty} \lim_{\beta\to\infty} T_{n,\beta} = T$ almost surely.*

The proof's core argument (given in Appendix B) is based on demonstrating that the quantities $r^\top \mathbf{U}_{\beta_{i,\cdot}}$ and $r^\top \mathbf{S}_{\beta_{i,\cdot}}$ converge to $r_{M(i)}$ and $r_{N(i)}$ respectively as the inverse temperature parameter $\beta$ approaches infinity. Once this convergence is established, the remainder of the proof follows easily from Theorems 5 and 6 in Azadkia & Chatterjee (2021), outlined in Appendix A.

---

[3]Even if applicable, FOCI is often not well-suited for deep learning applications, as shown in Sec. 5.2.
[4]Throughout the experiments, we use $\lambda = \max(1e^{10}, \max_{i,j} \mathbf{M}_{i,j} + \epsilon)$.

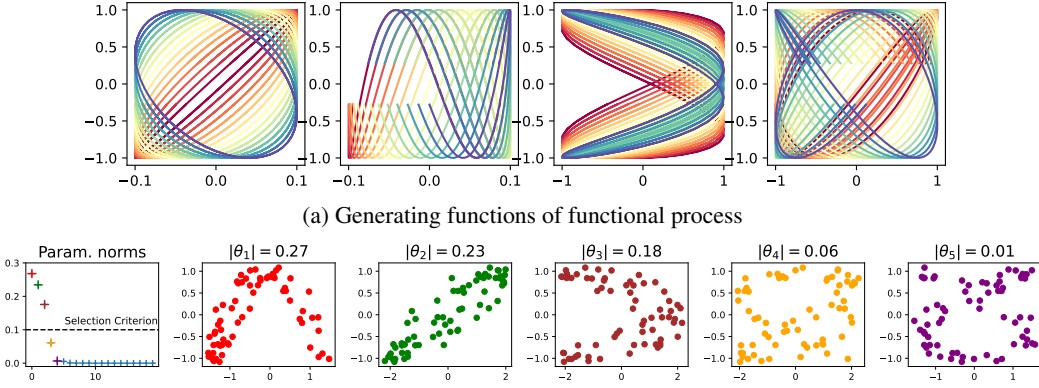

(a) Generating functions of functional process

(b) First plot: norms of $\theta$. Remaining plots: features with 5 largest param. norms (only first 3 selected).

Figure 1: Synthetic dataset experiment, detailed in Sec. 3.2. Out of 240 total features, our *vec*-(**dF1**) selects three informative, yet diverse features (corresponding to norms $0.27$, $0.23$, and $0.18$).

Making the estimator differentiable allows us to use $T_{n,\beta}$ in various ways. Considering the predictors $\mathbf{X}$, response variable $Y$ and potentially available sensitive attributes $\mathbf{X}_S$ or group affiliations $\mathbf{X}_G$, parameterization $f_\theta$, we highlight three ways to use $T_{n,\beta}$:

(**dF1**) $T_{n,\beta}(Y, f_\theta(\mathbf{X}))$: as a maximization objective, learning features that preserve ranks in the same fashion as the response

(**dF2**) $\ell(Y, \hat{Y}) + \lambda T_{n,\beta}(\mathbf{X_G}, f_\theta(\mathbf{X}))$: as a regularizer, penalizing the outputs (or learned features) $f_\theta(\mathbf{X})$ for being dependent on the protected groups $\mathbf{X}_G$, where $\ell$ denotes the standard loss used in machine learning

(**dF3**) $T_{n,\beta}(Y, f_\theta(\mathbf{X})|\mathbf{X}_S)$: as a conditioning objective, allowing to learn features that contain information about the response only after conditioning out the sensitive information $\mathbf{X}_S$

For instance, (**dF1**) can be utilized for feature selection or dimensionality reduction techniques. (**dF2**) can be employed to prevent the network from relying on spurious correlations when group attributes are available. (**dF3**) can be applied in fairness scenarios where we aim to avoid predictions based on certain personal information.

The remaining task is to select the parameterization $f_\theta(\cdot)$. In the following sections, we will focus on two options: *(i) vec* - a dot product parameterization $f_\theta(\mathbf{X}) = \theta \odot \mathbf{X}$, or *(ii) NN* - a neural network parameterization, $f_\theta(\cdot)$[5],[6]. Algorithm 2 provides a general outline for using the $T_{n,\beta}$ with a chosen parameterization, and specific instances of the algorithm are given in Appendix G.

---

**Algorithm 2** Differentiable FOCI (difFOCI)

---

**Input:** predictor $\mathbf{Z} \in \mathbb{R}^{n,p}$, response $Y \in \mathbb{R}^n$, and optional $\mathbf{X} \in \{\emptyset, S, G\}$, for sensitive $S \in \mathbb{R}^{n,d}$ or group info. $G \in \mathbb{R}^{n,d}$, $d \geq 1$
**Input:** parameterization $f_\theta \in \{vec, NN\}$, objective choice $T_{n,\beta} \in \{(\textbf{dF1}), (\textbf{dF2}), (\textbf{dF3})\}$
Initialize $\theta$
**for** $t = 1, ..., n_{\text{iter}}$ **do**
    $\mathcal{L} \leftarrow T_{n,\beta}(Y, f_{\theta_t}(\mathbf{Z})|\mathbf{X})$                                    // Applying difFOCI
    Update $\theta_{t+1} \leftarrow \text{Optim}(\mathcal{L}, \theta_t)$                           // Parameter update
**Output:** parameterization $f_\theta$

---

We proceed by testing whether difFOCI performs well at FOCI's main application - feature selection. We begin with a simulated dataset, followed by three experiments with increasing complexity.

---

[5]For example, with *vec*-(**dF1**) we denote using (**dF1**) with vector parameterization.
[6]We also tried *vec-NN* parameterization $f_\theta(\mathbf{X}) = \theta_2 \odot f_{\theta_1}(\mathbf{X})$, with $\theta = \{\theta_1, \theta_2\}$ but it did not show any improvements over the *NN* parameterization.

## 3.2 PRELIMINARY SYNTHETIC STUDY

To evaluate the feature selection performance of difFOCI, we utilize *vec*-(**dF1**) to obtain the objective $T_{n,\beta}(Y, \theta \odot \mathbf{X})$. Unlike FOCI, which returns a binary vector indicating whether a feature is selected or rejected, difFOCI's version *vec*-(**dF1**) yields a real-valued vector with components $(\theta_i)_{i\in[p]}$ representing the predictive informativeness of each corresponding feature (which allows taking into account feature variability). To perform feature selection, we need to choose a cutoff parameter $\upsilon$ and select the features with $|\theta_i| \geq \upsilon$.

Alg. 2 therefore requires the following hyperparameters: softmax temperature $\beta$, cutoff value $\upsilon$, and optimization parameters (e.g., learning rate $\gamma$, weight decay $\lambda$, minibatch size $b$, etc.). Our experimental analyses show that $\beta = 5$ and $\upsilon = 0.1$ yield consistently good performance, so we set these as fixed[7]. As a result, our algorithm simplifies solely to the hyperparameters used in conventional optimization methods, which are in Appendix G for all experiments.

**Environment.** As an initial example, we consider a data-generation process ideal for FOCI: from a large pool of features, a handful is sufficient for strong performance with $n \sim p$. The functional process is illustrated in Fig. 1a, crafted to generate a diverse set of features: *informative ones*, such as straight lines, sinusoids, or parabolas, and functions *individually uninformative, yet informative in multidimensional contexts*, e.g., ellipses, rotated parabolas, and more involved curves. This process includes 60 functions, each noised four times, resulting in $p = 240$ features with $n = 100$ points. Ideally, a feature selection method should pinpoint a small but diverse set of features[8].

**Baselines.** For comparative analysis, we employ various feature selection techniques from the scikit-learn library (Pedregosa et al., 2011). These include: *GenericUnivariateSelect* (GUS) for univariate feature selection, *SelectPercentile* (S.Per.), retaining only the top user-specified percentile of features, and statistical test-based methods: *SelectFpr* (FPR), *SelectFdr* (FDR), and *SelectFwe* (FWE) addressing false positive rate, false discovery rate, and family-wise error, respectively. Additionally, we employ *SelectKBest* (K.B) to select the best 25%, 50%, or 75% of features based on the ANOVA F-value test (Girden, 1992). We also benchmark against dimensionality reduction techniques including Linear Discriminant Analysis (LDA, Fisher (1936)), Principal Component Analysis (PCA, Wold et al. (1987)), and Uniform Manifold Approximation and Projection (UMAP, McInnes et al. (2018)), retaining 25%, 50%, and 75% of the features/principal components.

Throughout this and Sec. 4, we measure the performance by looking at the test error using Support Vector Regression (SVR, $C = 1.0$, $\epsilon = 0.2$) (Vapnik, 1963; 1964; Smola & Schölkopf, 2004). For SelectKBest, PCA, and UMAP, instead of reporting for $25, 50$, and $75\%$ of features/components separately, we only provide the results yielding the lowest mean-squared test error.

---

[7]A further discussion on this can be found in Appendix J

[8]The exact data-generating process is given in Appendix D.1

---

Table 1: Feature selection benchmark results in terms of test MSE. Our algorithms consistently yield the most accurate predictions while selecting one of the smallest feature subsets (as seen in (1a)). With $\hat{\mu}_y$, we denote predicting the overall mean and with *Full*, regressing to the whole dataset.

|  | GUS | S.Per. | FPR | FDR | FWE | K.B. 2 | K.B. 50 | K.B. 75 | FOCI | *vec*-(**dF1**) | *NN*-(**dF1**) |
|---|---|---|---|---|---|---|---|---|---|---|---|
| # Feat. Select. | 1 | 24 | 112 | 95 | 53 | 2* | 50* | 100* | 6 | 2 | N/A |
| Test MSE | 0.086 | 0.028 | 0.027 | 0.028 | 0.030 | 0.084 | 0.030 | 0.028 | 0.030 | 0.016 ± 0.02 | **0.012 ± 0.01** |

(a) Results from simulated data study, detailed in Sec. 3.2. Both (**dF1**) versions successfully inherit FOCI's strengths: they select a small number of features while exhibiting solid performance.

|  | $\hat{\mu}_y$ | Full | GUS | S.Per. | FPR | FDR | FWE | K.B. | UMAP | PCA | FOCI | *vec*-(**dF1**) | *NN*-(**dF1**) |
|---|---|---|---|---|---|---|---|---|---|---|---|---|---|
| Exp 1. | 1.38 | 0.22 | 0.93 | 0.94 | 0.53 | 0.54 | 0.68 | 0.54 | 1.14 | 1.02 | 0.21 | **0.02 ± 0.00** | 0.08 ± 0.01 |
| Exp 2. | 0.49 | 0.58 | 0.53 | 0.58 | 0.58 | 0.59 | 0.58 | 0.58 | 0.55 | 0.52 | 0.53 | 0.24 ± 0.00 | **0.02 ± 0.01** |
| Exp 3. | 0.35 | 0.31 | 0.32 | 0.32 | 0.34 | 0.34 | 0.33 | 0.33 | 0.33 | 0.34 | 0.30 | 0.23 ± 0.00 | **0.18 ± 0.01** |

(b) Results from three toy experiments, described in Sec. 4, show that both versions of (**dF1**) enhance FOCI's strengths. In Experiments 2 and 3, they are the only methods that outperform regressing to the mean $\hat{\mu}_y$.

Table 2: Feat. selection and dim. reduction benchmarks in terms of logistic test loss. Reported are the mean and std. across five random seeds. Our algorithms yield competitive predictions.

| | GUS | S.Per. | FPR | FDR | FWE | K.B. | UMAP | LDA | PCA | FOCI | *vec*-(**dF1**) | *NN*-(**dF1**) |
|---|---|---|---|---|---|---|---|---|---|---|---|---|
| Spambase | 10.70 | 6.05 | 2.92 | 2.92 | 2.92 | 3.39 | 2.97 | 3.20 | 3.12 | 3.04 | **2.56 ± 0.13** | **2.57 ± 0.19** |
| Toxicity | 14.41 | 12.98 | 17.30 | 12.98 | 18.02 | **10.09** | 12.98 | 15.86 | **10.00** | 16.30 | 11.61 ± 0.80 | **9.61 ± 1.50** |
| QSAR | 2.88 | 3.16 | 3.16 | 3.76 | 2.92 | 3.52 | 2.32 | **2.16** | **2.16** | 3.44 | 2.54 ± 0.07 | **2.11 ± 0.11** |
| Breast Cancer | 4.66 | 1.69 | 0.42 | 0.42 | 0.42 | **0.00** | 2.48 | 1.42 | 1.24 | 0.62 | **0.00 ± 0.00** | **0.00 ± 0.00** |
| Religious | 0.84 | 0.56 | 0.65 | 0.57 | 0.56 | 0.48 | 6.63 | 1.61 | 0.60 | 0.53 | **0.48 ± 0.03** | 0.56 ± 0.04 |

**Results.** Our approach selects a small, diverse, and informative set of features, resulting in good performance and showcasing successful inheritance of FOCI's main strengths (see Table 1a). The norms of the selection parameter $\theta$ are shown in Fig. 1b, demonstrating the evident relationship between the predictive informativeness of the features and the corresponding parameter norms.

We have discussed the recent advances and methodologies necessary to introduce difFOCI, as well as provided experimental analysis on a synthetic examples. We now proceed to more challenging examples, and ultimately to real-world datasets.

## 4 FROM FEATURE SELECTION TO FEATURE LEARNING

With a high-level understanding of difFOCI in place, we continue to assess its performance. We begin by highlighting two key observations we encountered during our preliminary experiments. We consider the following toy example: $Y = \sin(X_1) + 2\sin(X_2) + 3\sin(X_3) + \epsilon$, where $\epsilon_i \sim \mathrm{N}(0, \sigma_\epsilon^2)$, $i \in [n]$, and $\mathbf{X} \sim N(0, \sigma_x^2 \mathrm{I}_p)$, with $n = 2000$, $p = 10$, $\sigma_x = \sigma_\epsilon = 0.1$.

**Observation 1.**(†) The objectives (2) and (3) consistently capture the correct feature functional forms. Specifically, the values *(i)* $T_n\left(Y, \left[\sum_{i=1}^2 \sin(\pi_i^j(\mathbf{X}_{1:3}), \sin(\pi_3^j(\mathbf{X}_{1:3}))\right]\right)$, $j \in [3]$, *(ii)* $T_n(Y, \sin(\mathbf{X}_{1:3}))$, and *(iii)* $T_n(Y, \sum_{i=1}^3 \sin(\mathbf{X}_i))$ are all significantly greater than *(i)* $T_n\left(Y, \left[\sum_{i=1}^2 \pi_i^j(\mathbf{X}_{1:3}), \pi_3^j(\mathbf{X}_{1:3})\right]\right)$, *(ii)* $T_n(Y, \mathbf{X}_{1:3})$, and *(iii)* $T_n(Y, \sum_{i=1}^3 \mathbf{X}_i)$ (as illustrated in Figure 2a in the Appendix). Therefore, a more complex parameterization (than $f_\theta(\mathbf{X}) = \theta \odot \mathbf{X}$) might learn a nonlinear transformation of the features, maintaining ranks in a manner more consistent with the true functional forms.

**Observation 2.**(‡) The objectives (2) and (3) consistently prefer correct, lower-dimensional bases of the features. Specifically, $T_n(Y, \sum_{i=1}^3 \sin(\mathbf{X}_i))$ remains consistently greater than $T_n(Y, \sin(\mathbf{X}_{1:3}))$. Therefore, a more elaborate parameterization could learn an appropriate, possibly lower dimensional, basis transformation.

Motivated by these observations, we propose *NN* parameterizations to further explore the capabilities of difFOCI. We begin with simple one-hidden-layer Multi-layer Perceptrons (MLPs) as $f_\theta$ parameterizations. We set the output dimension to match the input, as this performed well across all experiments, though treating it as a hyperparameter might further enhance performance.

### 4.1 INITIAL ASSESSMENTS OF (**dF1**)

We now evaluate both *vec*-(**dF1**) and *NN*-(**dF1**) across three progressively challenging examples. We note that across all examples, FOCI selects the correct subset of the features more than 95 percent of the time. We set $p = 10$ throughout the experiments, and both $\sigma_\epsilon = \sigma_x = 0.1$. Full experimental details are given in Appendix F.

**Toy example 1: difFOCI successfully accounts for feature variability.** Here, we test whether *vec*-(**dF1**) and *NN*-(**dF1**) on the following example, previously introduced in Sec. 3: $Y = \sin(X_1) + 2\sin(X_2) + 3\sin(X_3) + \epsilon$, where $\epsilon_i \sim \mathrm{N}(0, \sigma_\epsilon^2)$, $i \in [n]$, and $\mathbf{X} \sim N(0, \sigma_x^2 \mathrm{I}_p)$, with $n = 2000$. In Table 1b, we observe that *vec*-(**dF1**) and *NN*-(**dF1**) successfully pinpoint the correct feature subset and account for feature variability, resulting in improved performance to that of FOCI. We expand on this in Appendix, Fig. 2b for *vec*-(**dF1**), where we can observe the correct proportionality of the coefficients in the regression equation and the learned parameters $\theta_1, \theta_2$ and $\theta_3$.[9]

---

[9] Note that this is already an improvement to FOCI, as it cannot take into account feature variability.

**Toy example 2: difFOCI can learn appropriate basis transformations.** The goal of this toy example is to examine whether *NN*-**(dF1)** effectively learns basis transformations. Data are generated as follows: $Y = \sin(X_1 + 2X_2 + 3X_3) + \epsilon$, where $\epsilon_i \sim \mathrm{N}(0, \sigma_\epsilon^2)$, $i \in [n]$, and $\mathbf{X} \sim N(0, \sigma_x^2 \, \mathrm{I}_p)$, with $n = 2000$. We affirmatively demonstrate its efficacy by examining the test loss after fitting the SVR - the substantially lower test error can be observed in Table 1b.

**Toy example 3: difFOCI simultaneously addresses mutual interactions, basis, and nonlinear transformations.** Our final example seeks to explore the full capabilities of **(dF1)** with NN parameterization, examining whether it can simultaneously discern complex, interrelated relationships as well as multiple transformations, encompassing both nonlinear and basis transformations. The data generation process is as follows: $Y = \sin\big((X_1 X_2)^2 + (X_2 X_3)^2 + (X_1 X_3)^2\big) + \epsilon$, where $\epsilon_i \sim \mathrm{N}(0, \sigma_\epsilon^2)$, $i \in [n]$, and $\mathbf{X} \sim N(0, \sigma_x^2 \, \mathrm{I}_p)$, with $n = 5000$. As evidenced in Table 1b (using a two-hidden-layer MLP[10]), we successfully learn effective transformations that result in strong performance.

**Summary.**  Throughout the experiments, both *vec*-**(dF1)** and *NN*-**(dF1)** yield strong performance, as seen in Table 1b. The two penultimate examples emphasize the potential capabilities of difFOCI; not only can it correctly identify the relevant subsets, but it also learns useful transformation, yielding the only method that outperforms random guessing (see $\hat{\mu}_y$ column in Table 1b).

## 5 EXPERIMENTS

Having examined **(dF1)** on synthetic problems and toy datasets, we now proceed to real-world datasets. We attempt to demonstrate the flexibility of difFOCI and highlight the promising potential of all three objectives: **(dF1)**-**(dF3)**. Our aim in this section is not solely to outperform existing benchmarks, but rather to showcase difFOCI's broad applicability, inspire further investigation into these applications, and explorations of new areas where the method can be applied.

### 5.1 REAL-WORLD DATA

In this section, we compare *vec*-**(dF1)** and *NN*-**(dF1)** to feature selection and dimensionality reduction methods using real-world datasets.

**Environments.** We evaluate our methods on five UCI datasets (Dua & Graff, 2017): Breast Cancer Wisconsin (Street et al., 1993), involving benign/malignant cancer prediction; Toxicity (Gul et al., 2021), aimed at determining the toxicity of molecules affecting circadian rhythms; Spambase (Hopkins et al., 1999), classifying emails as spam or not; QSAR (Ballabio et al., 2019), a set containing molecular fingerprints used for chemical toxicity classification, and Religious (Sah & Fokoué, 2019), aimed at identifying the source of religious books texts. We perform Logistic Regression (Cox, 1958) with default scikit-learn (Pedregosa et al., 2011) parameters (tol $=1\mathrm{e}{-}4$, $C = 1.0$). Dataset information is provided in Appendix C.

**difFOCI is competitive in feature selection and dimensionality reduction.**  As seen in Table 2 difFOCI achieves solid performance in the experiments. For *NN*-**(dF1)**, we use two-hidden-layer MLPs. The findings, which employ logistic loss, demonstrate that taking into account feature variability and using parameterization are crucial for improved performance compared to FOCI.

### 5.2 DOMAIN SHIFT/SPURIOUS CORRELATIONS

Here, we investigate an application of difFOCI to deep learning in the form of *NN*-**(dF2)**. The data consists of triplets $(Y, \mathbf{X}, \mathbf{X}_G)$, denoting the predictor, response variables, and group attributes, respectively. In this context, difFOCI can be employed as a regularizer to enforce the learning of uncorrelated features with respect to spurious attributes, thereby mitigating relying on spurious correlations and shortcuts in the model (Kenney, 1982).

---

[10]For this example, we found one-hidden-layer MLP not to be expressive enough.

Table 4: Worst group accuracy across several datasets. difFOCI obtains competitive performance.

| Dataset | difFOCI+ERM | difFOCI+DRO | ERM | DRO | JTT | Mixup | IRM |
|---|---|---|---|---|---|---|---|
| MultiNLI | **77.6 ± 0.1** | **77.5 ± 0.2** | 66.9 ± 0.5 | 77.0 ± 0.1 | 69.6 ± 0.1 | 69.5 ± 0.4 | 66.5 ± 1.0 |
| CivilComments | 66.32 ± 0.2 | **70.3 ± 0.2** | 64.1 ± 1.1 | **70.2 ± 0.8** | 64.0 ± 1.1 | 65.1 ± 0.9 | 63.2 ± 0.5 |
| CelebA | **89.32 ± 0.4** | **89.8 ± 0.9** | 65.0 ± 2.5 | **88.8 ± 0.6** | 70.3 ± 0.5 | 57.6 ± 0.5 | 63.1 ± 1.7 |
| NICO++ | **47.10 ± 0.7** | 46.3 ± 0.2 | 39.3 ± 2.0 | 38.3 ± 1.2 | 40.0 ± 0.0 | 43.1 ± 0.7 | 40.0 ± 0.0 |
| MetaShift | 83.10 ± 0.5 | **91.7 ± 0.2** | 80.9 ± 0.3 | 86.2 ± 0.6 | 82.6 ± 0.6 | 80.9 ± 0.8 | 84.0 ± 0.4 |
| CheXpert | 54.42 ± 3.2 | **75.3 ± 0.3** | 50.1 ± 3.5 | 73.9 ± 0.4 | 61.5 ± 4.3 | 40.2 ± 4.1 | 35.1 ± 1.2 |

**Environment.** We use Waterbirds dataset (Sagawa et al., 2019), which combines bird photographs from the Caltech-UCSD Birds-200-2011 dataset (Wah et al., 2011) with image backgrounds from the Places dataset (Zhou et al., 2017). The labels $Y = \{\text{waterbirds}, \text{landbirds}\}$ are placed against $G = \{\text{water}, \text{land}\}$ backgrounds, with waterbirds (landbirds) more frequently appearing against a water (land) background (exact details given in Table 7, Appx. E). Due to this spurious correlation, (Sagawa et al., 2019; Idrissi et al., 2022; Bell et al., 2024) observed that NNs (i.e., ResNet-50 (He et al., 2016), pre-trained on ImageNet (Deng et al., 2009)) tend to rely on the background to infer the label, rather than solely focusing on birds.

Table 3: Average and worst group accuracies for the Waterbirds dataset. We compare to the ERM and DRO, where e.s. stands for early-stopping and $l2$ for Ridge regularization. We can see that difFOCI performs comparably to state-of-the-art spurious correlation methods.

| | Average acc. | | Worst group acc. | |
|---|---|---|---|---|
| | Train | Test | Train | Test |
| ERM | 100 | **97.3** | 100 | 60.0 |
| ERM (e.s. + strong $l2$) | 97.6 | 95.7 | 35.7 | 21.3 |
| ERM + FOCI | 99.9 | 77.8 | 1.1 | 0.0 |
| ERM + *NN*-(**dF2**) | 99.9 | 93.7 | 92.0 | **85.7** |
| DRO | 100.0 | **97.4** | 100.0 | 76.9 |
| DRO (e.s. + strong $l2$) | 99.1 | 96.6 | 74.2 | 86.0 |
| DRO + FOCI | 99.5 | 74.5 | 6.1 | 3.9 |
| DRO + *NN*-(**dF2**) | 80.1 | 93.5 | 99.2 | **87.2** |

**Preventing reliance on spurious correlations.** We investigate the potential benefits of employing *NN*-(**dF2**) as a regularization technique, which penalizes the reliance of extracted features $f_{FE_\theta}$ on the spurious attribute $\mathbf{X}_G$ (i.e., the background) via $T_{n,\beta}(\mathbf{X}_G, f_{FE_\theta}(\mathbf{X}) \mid \mathbf{X}_G)$. From Tables 3-4, we can see that *NN*-(**dF2**) (applied to both ERM and DRO) compares competitively to state-of-the-art methods. The exact algorithm is given in 4. Experimental details, reported average accuracy and further examples are in Appendix G.

**difFOCI increases worst group accuracy while maintaining solid performance.** We can observe in Table 3 that *NN*-(**dF2**) successfully prevents the network from relying on the spuriously correlated background while improving worst group accuracy for both ERM and DRO. Apart from Waterbirds dataset, we also tested difFOCI on 5 additional datasets: two text datasets: MultiNLI (Williams et al., 2017), CivilComments (Borkan et al., 2019), and four image datasets: NICO++ (Zhang et al., 2023), CelebA (Liang & Zou, 2022), MetaShift (Liang & Zou, 2022) and CheXpert (Irvin et al., 2019). Full experimental details (including average accuracy performance) can be found in Appendix G. We experimented with various architectures: in addition to the ResNet-50, we used BERT and ViT-B with pretraining strategies like DINO and CLIP. Furthermore, we compared to Just Train Twice (Liu et al., 2021), Mixup (Zhang, 2017), and Invariant Risk Minimization (Arjovsky et al., 2019) as baselines. As shown in Table 5, difFOCI demonstrates competitive performance in terms of both average and worst-group accuracy.

## 5.3 FAIRNESS STUDY

Finally, we explore *NN*-(**dF3**). This section, while not the primary focus of our contribution, offers a complementary illustration of the difFOCI objective's versatility through a heuristic example. We found that this form (**dF3**) preserves the performance of the chosen parameterization while significantly reducing its predictivity of the sensitive attribute.

**Environments.** We utilize classification datasets with interpretable features and sensitive attributes: *(i)* Student dataset (Cortez & Silva, 2008), aimed at predicting if a student's performance surpasses a specific threshold (sex as the sensitive); *(ii)* Bank Marketing dataset (Moro et al., 2014) with predicting if a customer subscribes to a bank product (marital status as the sensitive); and two ACS

Table 5: *NN*-(**dF3**) allows preserving predictivity of $y$ while significantly reducing predictivity of $X_s$.

| Dataset | Features | Train acc: $y$ | Val. Acc: $y$ | Test acc: $y$ | Train acc: $X_s$ | Val. Acc: $X_s$ | Test acc: $X_s$ |
|---|---|---|---|---|---|---|---|
| Bank marketing | Stand. data | $91.32 \pm 2.3$ | $93.27 \pm 1.2$ | $90.05 \pm 2.0$ | $89.09 \pm 1.2$ | $72.26 \pm 1.5$ | $70.93 \pm 0.9$ |
| | (**dF3**) features | $90.81 \pm 1.8$ | $92.13 \pm 2.6$ | $89.35 \pm 1.1$ | $63.12 \pm 2.8$ | $62.24 \pm 0.7$ | $\mathbf{63.81 \pm 2.1}$ |
| Student data | Stand. data | $88.35 \pm 1.7$ | $79.63 \pm 0.9$ | $75.67 \pm 1.3$ | $95.68 \pm 2.1$ | $72.16 \pm 2.4$ | $71.21 \pm 1.5$ |
| | (**dF3**) features | $80.18 \pm 2.9$ | $72.16 \pm 1.6$ | $72.73 \pm 1.7$ | $59.47 \pm 1.1$ | $58.95 \pm 1.0$ | $\mathbf{48.89 \pm 1.1}$ |
| ASCI Income | Stand. data | $83.49 \pm 2.4$ | $85.10 \pm 2.1$ | $81.30 \pm 2.7$ | $68.97 \pm 1.6$ | $67.67 \pm 2.6$ | $66.00 \pm 0.7$ |
| | (**dF3**) features | $82.80 \pm 0.8$ | $81.99 \pm 1.5$ | $82.95 \pm 0.9$ | $56.58 \pm 1.2$ | $55.01 \pm 2.0$ | $\mathbf{52.73 \pm 2.0}$ |

datasets (Ding et al., 2021), *(iii)* Employment and *(iv)* Income, for predicting individual's employment status and whether their income exceeds a threshold, with sex and race as sensitive attributes in both datasets. Exact experimental details are provided in Appendix I.

**Findings.** Leveraging the conditional dependence expression in (3), our method flexibly incorporates sensitive features to facilitate fairer classification without exploiting sensitive data. Using NN-(**dF3**), we optimize $T_{n,\beta}(Y, f_\theta(\mathbf{X}) \mid \mathbf{X}_s)$ to learn features that are informative about $Y$, offering an optimization that heuristically seems to favor solutions less predictive of $\mathbf{X}_s$. Specifically, we train two NNs to predict $y$: the first NN was trained on $X$ (without $X_s$), while the second NN was trained on features $f_\theta(X)$ obtained using (**dF3**). We then used the final layers of both NNs to predict the sensitive $X_s$. As can be observed from Table 5, difFOCI (**dF3**) significantly reduces the predictability of $X_s$ (sometimes to chance level) without significantly impacting accuracy on $y$ - in some cases even slightly improves it.

**Despite conditioning out sensitive information, difFOCI delivers solid performance.** From Table 5, we see that *vec*-(**dF3**) demonstrates strong performance by effectively debiasing the network (as it cannot predict the sensitive $X_s$ well), while keeping informativeness regarding $y$. In Appendix I, we conduct another experiment with similar findings showcasing the promising potential of (**dF3**).

## 6 CONCLUSION

In this paper, we discussed two recent advancements in rank-based measures of correlation, critically examining the proposed estimators, including the FOCI algorithm and its barriers to adoption in machine learning. Leveraging these advancements, we introduced three enhanced and more adaptable versions of FOCI. We conducted several studies to showcase the retention of FOCI's strengths and the improvement of its weaknesses. We evaluated difFOCI's capabilities from toy examples, where our method was the sole one exceeding random guessing, to comprehensive real-world datasets involving feature selection and spurious correlations, where it demonstrated state-of-the-art performance. Finally, we proposed a direct application of our algorithm in fairness research, showcasing that difFOCI successfully debiases neural networks on several datasets.

### ACKNOWLEDGEMENTS AND FUNDING

This work has received funding from the French government, managed by the National Research Agency (ANR), under the France 2030 program with the reference ANR-23-IACL-0008. We extend our thanks to Samuel Bell, João Maria Janeiro, Badr Youbi Idrissi, Theo Moutakanni, Stéphane d'Ascoli and Timothée Darcet for feedback and support. Finally, we also thank Carolyn Krol for extensive consultation and support throughout this project.

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

APPENDIX

The organization of the appendix is as follows:

- **Section A** provides foundational material necessary for understanding proof methodology.
- **Section B** presents the technical results that support the conclusions drawn in our work, particularly in relation to Theorem 4.
- **Section C** gives further insights into the Toy Experiment 1.
- **Section D** offers details regarding the feature selection datasets, which include both the synthetic dataset and the UCI datasets.
- **Section E** contains information about the Waterbirds dataset utilized for feature learning.
- **Section F** includes additional comments on the fairness experiments, which involve two UCI datasets and American Community Survey (ACS) data, made available through the Folktables package.
- **Section G** elaborates on the experimental analyses and the configuration of hyperparameters.
- **Section H** gives three concrete examples of the pseudocodes required for Alg. 2.
- **Section I** provides experimental details regarding experiments in Section 5.3.
- **Section J** gives empirical evidence for the choice of fixing the parameter $\beta = 0.2$ by analyzing the performance of difFOCI for different values of $\beta$.
- **Section K** presents experimental evaluation of robustness to domain shift for feature selection using difFOCI **(dF1)**.

## A  ADDITIONAL TECHNICAL BACKGROUND

### A.1  AUXILIARY RESULTS REQUIRED FOR THE PROOF OF THM. 4

In this section, we provide the required results for the proof of Theorem 4. We begin by providing the full forms of the estimator, including the case $p = 0$.

If $p \geq 1$, the estimate of $T$ reads as follows:

$$T_n = T_n(Y, \mathbf{Z} \mid \mathbf{X}) := \frac{\sum_{i=1}^n \left( \min \{r_i, r_{M(i)}\} - \min \{r_i, r_{N(i)}\} \right)}{\sum_{i=1}^n \left( r_i - \min \{r_i, r_{N(i)}\} \right)}. \tag{5}$$

And if $p = 0$, we obtain:

$$T_n = T_n(Y, \mathbf{Z}) := \frac{\sum_{i=1}^n \left( n \min \{r_i, r_{M(i)}\} - l_i^2 \right)}{\sum_{i=1}^n l_i (n - l_i)}, \tag{6}$$

with $M(i)$ denoting the index $j$ such that $\mathbf{Z}_j$ is the nearest neighbor of $\mathbf{Z}_i$, and $r_i$, $l_i$ as defined in Sec. 2.2.

We then proceed by defining:

1. $P_n(Y, \mathbf{X}) := \frac{1}{n^2} \sum_{i=1}^n (r_i - \min\{r_i, r_{N(i)}\})$, and
2. $Q_n(Y, \mathbf{Z}|\mathbf{X}) := \frac{1}{n^2} \sum_{i=1}^n (\min\{r_i, r_{M(i)}\} - \min\{r_i, r_{N(i)}\})$.

These two quantities are important for the following theorems:

**Theorem 5.** *(Azadkia & Chatterjee, 2021) Suppose that $p \geq 1$. As $n \to \infty$, the statistics $Q_n(Y, \mathbf{Z} \mid \mathbf{X})$ and $P_n(Y, \mathbf{X})$ converge almost surely to deterministic limits. Call these limits a and b, respectively. Then*

*1. $0 \leq a \leq b$.*

2. $Y$ is conditionally independent of $\mathbf{Z}$ given $\mathbf{X}$ if and only if $a = 0$.

3. $Y$ is conditionally a function of $\mathbf{Z}$ given $\mathbf{X}$ if and only if $a = b$.

4. $Y$ is not a function of $\mathbf{X}$ if and only if $b > 0$.

*Explicitly, the values of $a$ and $b$ are given by*

$$a = \int \mathbb{E}(\text{Var}(\mathbb{P}(Y \geq t \mid \mathbf{Z}, \mathbf{X}) \mid \mathbf{X}))d\mu(t)$$

*and*

$$b = \int \mathbb{E}\left(\text{Var}\left(1_{\{Y \geq t\}} \mid \mathbf{X}\right)\right) d\mu(t)$$
$$= \int \mathbb{E}(\mathbb{P}(Y \geq t \mid \mathbf{X})(1 - \mathbb{P}(Y \geq t \mid \mathbf{X})))d\mu(t).$$

Next, suppose that $p = 0$. Define $Q_n(Y, \mathbf{Z}) := \frac{1}{n^2} \sum_{i=1}^{n} \left(\min\{r_i, r_{M(i)}\} - \frac{L_i^2}{n}\right)$ and $P_n(Y) := \frac{1}{n^3} \sum_{i=1}^{n} L_i (n - L_i)$, where $L_i$ is the number of $j$ such that $Y_j \geq Y_i$.. Then, one can show the following:

**Theorem 6.** *(Azadkia & Chatterjee, 2021) As $n \to \infty$, $Q_n(Y, \mathbf{Z})$ and $P_n(Y)$ converge almost surely to deterministic limits $c$ and $d$, satisfying the following properties:*

1. $0 \leq c \leq d$.

2. $Y$ is independent of $\mathbf{Z}$ if and only if $c = 0$.

3. $Y$ is a function of $\mathbf{Z}$ if and only if $c = d$.

4. $d > 0$ if and only if $Y$ is not a constant.

*Explicitly,*

$$c = \int \text{Var}(\mathbb{P}(Y \geq t \mid \mathbf{Z}))d\mu(t),$$

*and*

$$d = \int \text{Var}\left(1_{\{Y \geq t\}}\right) d\mu(t) = \int \mathbb{P}(Y \geq t)(1 - \mathbb{P}(Y \geq t))d\mu(t).$$

The two aforementioned theorems serve as the key ingredients to Theorems 2 and 3, as well as the proof of Thm. 4, which is given in Sec. B.

## A.2 WORST-GROUP-ACCURACY (WGA) METHODS

Below, we mention the two most-popular methods appearing in the literature on WGA maximization:

**ERM.** Empirical Risk Minimization (ERM), proposed by Vapnik (2006), chooses the predictor minimizing the empirical risk $\frac{1}{n} \sum_{i=1}^{n} \ell(f(x_i), y_i)$. ERM does not use attribute (group) labels.

**Group DRO.** Group Distributionally Robust Optimization (gDRO) as proposed by Sagawa et al. (2019) aims to minimize the maximum loss across different groups. The objective is formulated as:

$$\sup_{q \in \Delta_{|G|}} \sum_{g=1}^{|G|} \frac{q_g}{n_g} \sum_{i=1}^{n_g} \ell(f(x_i), y_i),$$

where $G = Y \times A$ represents the set of all groups, $\Delta_{|G|}$ denotes the $|G|$-dimensional simplex, and $n_g$ is the number of examples belonging to group $g \in G$ within the dataset. As a result, gDRO incorporates attribute labels. Specifically, gDRO assigns a dynamic weight $q_g$ to the minimization of the empirical loss for each group, which is proportional to its current error rate.

**Other methods.** The body of literature on robust, worst-group optimization is rapidly expanding, making it infeasible to compare all available methods thoroughly. Additional examples of robust learners that do not utilize attribute information (like ERM) include Learning from Failure (Nam et al., 2020), the Too-Good-to-be-True prior (Dagaev et al., 2023), Spectral Decoupling (Pezeshki et al., 2021), Just-Train-Twice (Liu et al., 2021), and the George clustering algorithm (Sohoni et al., 2020). Conversely, methods that incorporate attribute information (like gDRO and difFOCI) include Conditional Value at Risk (Duchi et al., 2019), Predict then Interpolate (Bao et al., 2021), Invariant Risk Minimization (Arjovsky et al., 2019), and a wide range of domain-generalization algorithms (Gulrajani & Lopez-Paz, 2020).

## B  PROOF OF THEOREM 4.

In this section, we re-state Theorem 4 and prove it.

**Theorem 4.** Let $\beta \in \mathbb{R}^+$. Suppose that $Y$ is not almost surely equal to a measurable function of $\mathbf{X}$. Then, $\lim_{n \to \infty} \lim_{\beta \to \infty} T_{n,\beta} = T$ almost surely.

*Proof.* Let $Y$ be a random variable and $\mathbf{X} = (X_1, \ldots, X_p)$ and $\mathbf{Z} = (Z_1, \ldots, Z_q)$ be random vectors, defined on the same probability space. Here $q \geq 1$ and $p \geq 0$. The value $p = 0$ means that $\mathbf{X}$ does not have any components. By $\mu$, we denote the law of $Y$.

Recall that we denote with $r_i$ the rank of $Y_{(i)}$, i.e., the number of $j$ for which $Y_{(j)} \leq Y_{(i)}$, and with $l_i$ the number of $j$ such that $Y_{(j)} \geq Y_{(i)}$. For each index $i$, $N(i)$ is the index $j$ where $\mathbf{X}_j$ is the closest to $\mathbf{X}_i$, and $M(i)$ is the index $j$ where the pair $(\mathbf{X}_j, \mathbf{Z}_j)$ is closest to $(\mathbf{X}_i, \mathbf{Z}_i)$ in $\mathbb{R}^{p+q}$ w.r.t. the Euclidean metric and resolving ties randomly.

The two quantities $Q_n(Y, \mathbf{Z} \mid \mathbf{X})$ and $P_n(Y, \mathbf{X})$ and their respective limits $a$ and $b$ (see Theorems 5 and 6) are key to proving Theorems 2 and 3. In order to prove that $T_{n,\beta}$ converges to the same limit as $T_n$, we have to introduce the following two quantities:

1. $P_{n,\beta}(Y, \mathbf{X}) := \frac{1}{n^2} \sum_{i=1}^n \left( r_i - \min\{r_i, r^\top \mathbf{S}_{\beta_{i,\cdot}}\} \right)$, and

2. $Q_{n,\beta}(Y, \mathbf{Z}|\mathbf{X}) := \frac{1}{n^2} \sum_{i=1}^n \left( \min\{r_i, r^\top \mathbf{U}_{\beta_{i,\cdot}}\} - \min\{r_i, r^\top \mathbf{S}_{\beta_{i,\cdot}}\} \right)$,

with $\mathbf{S}_\beta = \sigma_\beta(-(\mathbf{M} + \lambda \mathbf{I}_n))$, $\mathbf{U}_\beta = \sigma_\beta(-(\hat{\mathbf{M}} + \lambda \mathbf{I_n}))$, and $\hat{M}_{i,j} = \|(\mathbf{X}_i, \mathbf{Z}_i) - (\mathbf{X}_j, \mathbf{Z}_j)\|$, with $\sigma_\beta$ the softmax function as defined in Sec. 3.1.

Now, define $\gamma_i := |r_i - r_{M(i)}|$ and $\delta_i := |r_i - r_{N(i)}|$. Let $\epsilon = \min(\gamma_1, \ldots, \gamma_n, \delta_1, \ldots, \delta_n)$. Then, by the continuity properties of $\sigma_\beta(\cdot)$ and setting $\lambda = \max(1e^{10}, \max_{i,j} \mathbf{M}_{i,j} + \epsilon)$, we have $\lim_{\beta \to \infty} \mathbf{S}_{\beta_{i,\cdot}} = \lim_{\beta \to \infty} \sigma_\beta(-(\mathbf{M} + \lambda \mathbf{I}_n))_{i,\cdot} = \mathbb{1}\{i = \arg\max_{j \in [n] \setminus i} -\|\mathbf{M}_{i,\cdot}\|\} = \mathbb{1}\{i = \arg\min_{j \in [n] \setminus i} \|\mathbf{X}_i - \mathbf{X}_j\|\} = N(i)$. One can similarly show that $\lim_{\beta \to \infty} \mathbf{U}_{\beta_{i,\cdot}} = \lim_{\beta \to \infty} \sigma_\beta(-(\hat{\mathbf{M}} + \lambda \mathbf{I}_n))_{i,\cdot} = M(i)$. Therefore, we can choose $n > N^* = \max(N_1, N_2)$, such that $\forall n > N_1, \max_i |r_{N(i)} - r^\top \mathbf{S}_{\beta_{i,\cdot}}| < \epsilon$, and $\forall n > N_2, \max_i |r_{M(i)} - r^\top \mathbf{U}_{\beta_{i,\cdot}}| < \epsilon$. Then, we can easily show that $Q_{n,\beta}(Y, \mathbf{Z}|\mathbf{X})$ converges to the limit $c$, with $c = \lim_{n \to \infty} Q_n(Y, \mathbf{Z}|\mathbf{X})$:

$$|Q_{n,\beta}(Y,\mathbf{Z}|\mathbf{X}) - c| = |\frac{1}{n^2}\sum_{i=1}^{n}(\min\{r_i, r^\top\mathbf{U}_{\beta_{i,.}}\}\} - \min\{r_i, r^\top\mathbf{S}_{\beta_{i,.}}\}\}) - c|$$

$$\leq |\frac{1}{n^2}\sum_{i=1}^{n}(\min\{r_i, r_{M(i)}\}\} - \min\{r_i, r_{N(i)}\}) - c| + \frac{2n\epsilon}{n^2}$$

$$\leq |Q_n(Y,\mathbf{Z}|\mathbf{X}) - c| + \frac{2n\epsilon}{n^2},$$

where both terms go to zero as we take $n$ to infinity (for the first term, see Thm. 5). One can also straightforwardly show that $P_{n,\beta}(Y,\mathbf{X})$ converges to the same limit $b$ as $P_n(Y,\mathbf{X})$.

Finally, we can closely follow Sec. 10 in Azadkia & Chatterjee (2021) to conclude; For case $p \geq 1$, we recall the quantities $a$ and $b$ from the statement of Theorem 5, and notice that $T = a/b$. By Theorem 5, $Q_n \to a$ and $S_n \to b$ in probability. Thus, $T_n \to a/b = T$ in probability. This proves Theorem 4 when $p \geq 1$. Finally, for case $p = 0$, here $T = c/d$, where $c$ and $d$ are the quantities from Theorem 6. Note that $T_n = Q_n/S_n$, where $Q_n = Q_n(Y,\mathbf{Z})$ and $S_n = S_n(Y)$. By Theorem 6, $Q_n \to c$ and $S_n \to d$ in probability. Thus, $T_n \to c/d = T$ in probability. This proves Theorem 4 when $p = 0$.

$\square$

## C  CONTINUATION OF TOY EXPERIMENT 1.

We present a plot for the two observations discussed in Section 5, as well as Toy Example 1. The left plot shows that the differences between the three observations are all statistically significant, while the right plot highlights two key strengths of our method: it quickly stabilizes, and the parameter norms reflect the variability of the features.

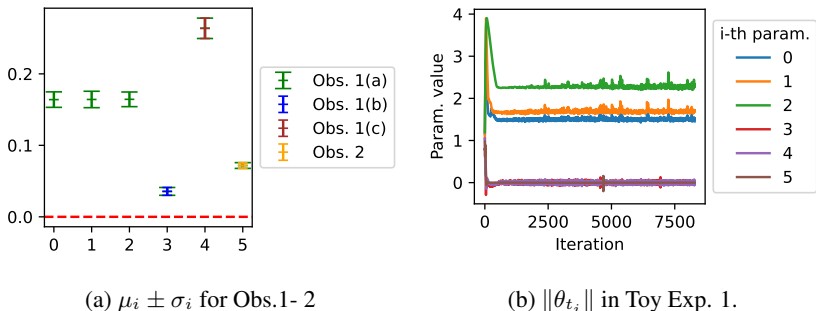

(a) $\mu_i \pm \sigma_i$ for Obs.1- 2    (b) $\|\theta_{t_i}\|$ in Toy Exp. 1.

Figure 2: Left: Mean and std. across 50 random inits. All expressions yield values significantly greater than zero. Right: Development of the first five parameters in Toy Exp 1.

## D  EXPERIMENTAL ENVIRONMENTS

In the following, we provide details on the environments used in our experiments in Section 5. We list the number of features, samples, and classes in each UCI environment in Table 6.

|  | Spambase | Toxicity | QSAR | Breast Cancer | Religious |
|---|---|---|---|---|---|
| $n$ | 4601 | 171 | 8992 | 569 | 8265 |
| $p$ | 57 | 1203 | 1024 | 30 | 590 |
| # classes | 2 | 2 | 2 | 2 | 8 |

Table 6: Feature Selection Dataset details

## D.1 SYNTHETIC ENVIRONMENT

Here, we briefly describe how we generate the synthetic environment depicted in Fig. 1, the synthetic dataset that is created using trigonometric transformations and permutations of parameters.

Let $x$ be a linearly spaced vector defined as $x = \text{linspace}(-6, 6, 100)$. Define the parameters $a$, $b$, and $c$ as: $a = \text{linspace}(0.1, 2, 4)$, $b = \text{linspace}(0.1, 2, 15)$, and $c = \text{linspace}(-1, 1, 4)$, where linspace(a, b, n) represents $n$ uniformly spaced points in the interval $[a, b]$. Features are then generated using the formula:

$$f(x)_{a,b,c} = a \cdot \sin(b \cdot x + c) \tag{7}$$

where $a$, $b$, and $c$ are elements from the Cartesian product of the parameter sets $a$, $b$, and $c$. The features are stored in a matrix $X$ where each column represents a feature vector. For each feature vector, transformations are applied as follows: $X_{\text{new}} = (-1)^{i+1} \cdot X[:, i \cdot 15 + j]$ for $i \in \{0, 1\}$ and $j \in \{0, \ldots, 14\}$. Additional transformations are applied based on a permutation of parameters $c$, $a$, and $b$. The transformed features are: $X_{\text{final}} = -1 \cdot X_{\text{new}}[:, i \cdot 15 + j]$ for selected indices $i$ and all $j$. The final dataset $X$ is obtained by concatenating all transformed features and adding Gaussian noise: $X = X + \mathcal{N}(0, 0.1)$ four times, yielding $n = 100$ and $p = 4 * 15 * 4 = 240$. The predictor variable is calculated as $y = \sin(x)$ - we do not add further noise here as the features already contain noise.

## D.2 UCI DATASETS - FEATURE SELECTION

Below, we briefly describe the five UCI datasets (Dua & Graff, 2017) used in our feature selection comparison.

### D.2.1 SPAMBASE

The "Spambase" dataset (Hopkins et al., 1999) is designed for classifying emails as spam or non-spam. It consists of 4,601 email instances with 57 features, characterized by both integer and real values. The dataset is multivariate and is often used in computer science, with classification as the primary task.

The dataset includes diverse types of spam, such as product ads, money schemes and chain letters. The goal is to identify whether an email is spam, with some non-spam indicators like the word "george" or area code "650" reflecting personalized filters.

### D.2.2 TOXICITY

The "Toxicity" dataset (Gul et al., 2021) contains data on 171 small molecules designed for the functional domains of CRY1, a core clock protein involved in circadian rhythm regulation. Of these molecules, 56 are toxic, while the rest are non-toxic. The dataset is tabular, with 1,203 molecular descriptors per instance. The primary task is classifying molecules as toxic or non-toxic.

### D.2.3 QSAR

The "QSAR Oral Toxicity" dataset (Ballabio et al., 2019) consists of 8,992 chemical compounds represented by 1,024 binary molecular fingerprint attributes. These attributes are used to classify the chemicals into two categories: very toxic (positive) or not very toxic (negative). The dataset is multivariate and is often used in physics and chemistry, with classification as the main associated task.

### D.2.4 BREAST CANCER

The "Breast Cancer Wisconsin (Diagnostic)" dataset (Street et al., 1993) is used for classifying breast cancer diagnoses based on data from fine needle aspirates (FNA) of breast masses. It consists of 569 instances with 30 real-valued features that describe characteristics of cell nuclei in digitized images. The dataset is multivariate and is often used in the field of health and medicine, with classification as the primary task. The features were created through an exhaustive search using the Multisurface Method-Tree and linear programming techniques to create a decision tree.

### D.2.5 RELIGIOUS

The dataset, "A Study of Asian Religious and Biblical Texts," (Sah & Fokoué, 2019) primarily consists of texts sourced from Project Gutenberg. It includes a collection of key religious and philosophical texts, such as the Upanishads, Yoga Sutras, Buddha Sutras, Tao Te Ching, and selections from the Bible (Books of Wisdom, Proverbs, Ecclesiastes, and Ecclesiasticus). The dataset is multivariate and is analyzed in Social Science contexts, with associated tasks including classification and clustering.

## E WATERBIRDS DATASET - FEATURE LEARNING

| Dataset | Target | Group Counts | | Class Counts |
| --- | --- | --- | --- | --- |
| | | Water | Land | |
| Waterbirds | Land bird | 56 | 1057 | 1113 |
| | Water bird | 3498 | 184 | 3682 |

Table 7: (Sub)group counts for the Waterbirds Dataset

The Waterbirds dataset consists of images of birds that have been digitally cut and pasted onto various backgrounds. The objective is to classify the specimens as either water birds or land birds. The group attribute indicates whether the bird is depicted in its natural habitat. The details of class counts are given in Tab. 7. While performing hyperaparameter search, each experiment is run on one Nvidia Tesla V100 GPU.

|  | Bank Marketing | Student Performance | ACS Employment | ACS Income |
|---|---|---|---|---|
| $n$ | 41,188 | 395 | 3,236,107 | 1,664,500 |
| $p$ | 20 | 30 | 17 | 10 |
| # classes | 2 | 2 | 2 | 2 |
| # Protected attributes | Marital Status | Sex | Race, Sex | Race, Sex |

Table 8: Number of samples, parameters, classes, and sensitives for each dataset

## F  FAIRNESS EXPERIMENTS

### F.1  UCI DATASETS - FAIRNESS

The UCI datasets (Dua & Graff, 2017) used in our fairness experiments are briefly described below.

#### F.1.1  BANK MARKETING

The "Bank Marketing" dataset (Moro et al., 2014) contains data from direct marketing campaigns (phone calls) conducted by a Portuguese bank. The goal is to classify whether a client will subscribe to a term deposit (variable 'y'). The dataset is multivariate, with 45,211 instances and 16 features that are either categorical or integer. The ratio of sensitive marital status is 60-30-10% (married, single, divorced).

The marketing campaigns often involved multiple contacts with the same client to determine if they would subscribe to the term deposit. This dataset is used in business applications, with classification being the main associated task.

#### F.1.2  STUDENT PERFORMANCE

The "Student Performance" dataset (Cortez & Silva, 2008) aims to predict the performance of secondary education (high school) students. It consists of 649 instances and 30 integer features, and the associated tasks include classification and regression.

The data collected from two Portuguese schools includes student grades, demographic, social, and school-related information. Two separate datasets cover performance in Mathematics and Portuguese language. The target variable, G3 (final grade, whether it is $\geq 12$ or not), is strongly correlated with G1 and G2 (grades from earlier periods), making it more challenging but useful to predict G3 without using G1 and G2, as we do in our experiments. This dataset supports educational performance modeling in the Social Science domain. The ratio of sensitive sex is 50-50%.

### F.2  FOLKTABLES DATASET - FAIRNESS

The two datasets below are taken from the Folktables package (Ding et al., 2021), designed to provide access to datasets derived from the US Census. It features a range of pre-defined prediction tasks across various domains, such as income, employment, health, transportation, and housing. Users can also create new prediction tasks within the US Census data ecosystem. Additionally, the package facilitates systematic studies on the impact of distribution shifts, allowing each prediction task to be applied to datasets covering multiple years and all states in the US. We use the Alabama data from 2018 with the 1 year horizon.

#### F.2.1  INCOME

The task is to predict whether an individual's income exceeds $50,000 based on a filtered sample of the ACS PUMS data. The sample includes individuals aged 16 and older who reported working at least 1 hour per week over the past year and earning a minimum income of $100. The $50,000 threshold was selected to make this dataset a potential replacement for the UCI Adult dataset (Kohavi & Becker, 1994), although the original paper provides additional datasets with different income thresholds, as detailed in their Appendix B. We use the California data from 2018 with the 1 year horizon. The ratio of sensitive for sex is 50-50% and for race 62%-17%-5% for White, Asian, Black (other minorities include American Indian, Hawaiian, etc.)

### F.2.2 EMPLOYMENT

The objective of this task is to predict whether an individual is employed, using a filtered sample from the ACS PUMS data. This sample has been carefully curated to include only those individuals who are between the ages of 16 and 90. The ratio of sensitive for sex is 50-50% and for race 62%-17%-5% for White, Asian, Black (other minorities include American Indian,Hawaiian, etc.)

Both tasks contain codes regarding the selected features in Sec. 5.3. The codes are explained below.

#### DEMOGRAPHIC VARIABLES

1. **OCCP**:
   (a) Person's occupation
   (b) approximately 500 categories (management, business, science, arts, service, sales, office, construction, maintenance, production, transportation, material moving, etc.)

2. **COW**:
   (a) Class of worker
   (b) 10 categories (e.g., employee of a private for-profit company, local government employee, state government employee, federal government employee, self-employed, working without pay, etc.)

3. **POBP**:
   (a) Place of birth
   (b) approximately 300 categories (countries/states of birth), range of values includes most countries and individual U.S. states

4. **SEX**:
   (a) range of values: Male and Female

5. **MAR**:
   (a) Person's marital status
   (b) 5 categories (married, widowed, divorced, separated, never married, or under 15 years old)

6. **ANC**:
   (a) Ancestry
   (b) 5 different categories (single, multiple, unclassified, not reported, suppressed information)

7. **AGEP**:
   (a) Age, Range of Values: 0-99

8. **ESP**:
   (a) Employment status of parents
   (b) 9 different categories (Living with two parents - both in labor force, living with two parents - father only in labor force, living with father - father in labor force, living with father - father not in labor force, etc.)

#### CITIZENSHIP AND MIGRATION

1. **CIT**:
   (a) Citizenship status
   (b) 5 categories (Born in the U.S., Born abroad of American parent(s), U.S. citizen by naturalization, Not a citizen of the U.S., Born in Puerto Rico, Guam, the U.S. Virgin Islands, or the Northern Marianas)

2. **MIG**:
   (a) Mobility status (whether the person lived at the same location 1 year ago)
   (b) 4 categories (N/A if less than 1 year old, Yes - same house, No - outside U.S. and Puerto Rico, No - different house in U.S. or Puerto Rico)

EDUCATION

1. **SCHL**:

    (a) Amount of schooling completed

    (b) 24 categories (No schooling completed, Nursery school/preschool, Kindergarten, Grade 1, Grade 2,..., Regular high school diploma, GED or alternative credential, Bachelor's degree, Master's degree, etc.)

RACE AND ETHNICITY

1. **NAT**:

    (a) Whether native or foreign born

    (b) 2 categories (native or not)

2. **RAC1P**:

    (a) Recorded detailed race code

    (b) 9 categories (White alone, Black or African American alone, American Indian alone, Alaska Native alone, Asian alone, Some Other Race alone, etc.)

3. **MIL**:

    (a) Military service

    (b) 5 Categories (Less than 17 years old, Now on active duty, On active duty in the past but not now, Only on active duty for training, Never served in the military)

DISABILITY AND SENSORY IMPAIRMENTS

1. **DIS**:

    (a) Disability recorded: With or without

2. **DEAR**:

    (a) Hearing difficulty: Yes or No

3. **DEYE**:

    (a) Vision difficulty: Yes or No

4. **DREM**:

    (a) Cognitive difficulty: Yes or No

For further explanation on the codes, we invite the reader to see Appendix B.1 and B.4 in the original paper (Ding et al., 2021). Below, in Table 9, we can see that conditioning on multiple sensitive attributes removes additional features, highlighting the potential of the *vec*-(**dF3**) method to examine interactions between several sensitive attributes, as well as several features simultaneously.

| Data | Race | Sex | Both |
|---|---|---|---|
| Employment | OCCP | COW | OCCP, COW, POB |
| Income | MAR, ANC | MAR, ANC, CIT, MIG | MAR, ANC, CIT, MIG, SCHL, NAT |

Table 9: ACS dataset features which were not selected when conditioned on race, sex or both, represented in first, second and last column, respectively.

| Dataset | Year | Size | N. Classes | Modality | Architecture |
|---------|------|------|------------|----------|--------------|
| MultiNLI | 2017 | 300k | 3 | Text | BERT |
| CivilComments | 2019 | 250k | 2 | Text | BERT |
| CelebA | 2015 | 200k | 2 | Image | ResNet-50 w. ImageNet |
| NICO++ | 2022 | 90k | 60 | Image | ViT-B w. DINO |
| MetaShift | 2022 | 3.5k | 2 | Image | VIT-B w. CLIP |

Table 10: Dataset Overview for experiments performed in Section 5.2.

| Dataset | difFOCI+ERM | difFOCI+DRO | ERM | DRO | JTT | Mixup | IRM |
|---------|-------------|-------------|-----|-----|-----|-------|-----|
| MultiNLI | $\mathbf{81.9 \pm 0.2}$ | $\mathbf{81.8 \pm 0.5}$ | $81.4 \pm 0.1$ | $80.2 \pm 0.6$ | $81.2 \pm 0.4$ | $80.7 \pm 0.1$ | $77.7 \pm 0.3$ |
| CivilComments | $\mathbf{86.3 \pm 0.1}$ | $81.9 \pm 0.3$ | $85.7 \pm 0.4$ | $82.3 \pm 0.4$ | $84.3 \pm 0.5$ | $84.9 \pm 0.3$ | $85.4 \pm 0.2$ |
| CelebA | $94.4 \pm 1.1$ | $92.9 \pm 2.1$ | $94.9 \pm 0.2$ | $93.1 \pm 0.6$ | $92.4 \pm 1.6$ | $\mathbf{95.7 \pm 0.2}$ | $94.5 \pm 1.0$ |
| NICO++ | $\mathbf{85.7 \pm 0.3}$ | $85.8 \pm 0.5$ | $84.7 \pm 0.6$ | $83.0 \pm 0.1$ | $85.3 \pm 0.1$ | $84.2 \pm 0.4$ | $84.7 \pm 0.5$ |
| MetaShift | $\mathbf{92.1 \pm 0.2}$ | $91.8 \pm 0.3$ | $91.3 \pm 0.5$ | $89.0 \pm 0.2$ | $90.7 \pm 0.2$ | $91.2 \pm 0.4$ | $\mathbf{91.5 \pm 0.6}$ |
| CheXpert | $87.1 \pm 0.3$ | $81.9 \pm 0.5$ | $86.5 \pm 0.3$ | $77.9 \pm 0.4$ | $75.7 \pm 1.7$ | $82.2 \pm 5.1$ | $\mathbf{90.0 \pm 0.2}$ |

Table 11: Average accuracy for benchmark methods on various datasets performed in Section 5.2. We can see that on almost all datasets, diFFOCI performs competitively.

## G EXPERIMENTAL CONFIGURATIONS

For the first two toy examples, we use a one-hidden-layer MLP with a configuration of 10-20-10 neurons. In contrast, the third example employs a two-hidden-layer MLP structured as 10-20-20-10 neurons. For all benchmarks using *vec*-(**dF1**) and *vec*-(**dF3**), we initialize the parameter $\theta$ from a $\theta \sim N(1, \sigma^2 I_p)$, with $\sigma^2 = 0.1$. In the case of *NN*-(**dF1**), we design a one-layer MLP where the hidden dimension is double that of the input layer, and the output layer has the same number of neurons as the input layer. The ReLU function is used as the activation function. All input data is standardized, and across all benchmarks, we perform a (75-15-10)% train-validation-test split. For the regression experiments (toy examples), we employ SVR ($C = 1.0$, $\epsilon = 0.2$) along with the Adam optimizer (Kingma & Ba, 2017). For classification (UCI, Bank Marketing, Student and ACS Datasets), we employ Logistic Regression (Cox, 1958). For the Waterbirds dataset, we train ResNet-50 models pre-trained on ImageNet (Deng et al., 2009) using the SGD optimizer with the PyTorch (Paszke et al., 2017) implementation of *BCEWithLogitsLoss*, which combines a Sigmoid layer and the BCELoss in one single class.

We adjust the learning rate and weight decay from the set $\{10^{-4}, 10^{-3}, 10^{-2}, 10^{-1}, 5^{-4}, 5^{-3}, 5^{-2}\}$[11]. The number of epochs is optimized within the range $\{10, 20, 50, 100\}$, and batch sizes are chosen from $\{8, 16, 32, 64, 128, 256, 512, 1024, 2048, 4096\}$. Notably, we train the Waterbirds dataset for 360 epochs, in line with previous research. As mentioned earlier, we keep the clipping parameter $\nu = 0.1$ and the softmax temperature parameter $\beta = 5$ consistent across all experiments. The value of $\eta$ for gDRO is set to 0.1. Each combination of hyperparameters is executed three times to compute the average and standard deviation of the chosen loss metric. We select the best models (in terms of hyperparameter combinations and epochs) based on the lowest MSE/logistic loss observed on the validation set, and for Waterbirds, we choose based on worst-group accuracy. We provide the hyperparameter configurations used to obtain our results in Table 13. .

Finally, for the experiments on NICO++ (Zhang et al., 2023), MultiNLI (Williams et al., 2017), CivilComments (Borkan et al., 2019) and CelebA (Liang & Zou, 2022) we follow the experimental configuration from Yang et al. (2023), who provided a very useful codebase for benchmarking various methods, which we are thankful for. We briefly describe these datasets below, where the hyperparameter joint distribution is taken directly from their codebase. For each algorithm, we perform a thorough hyperparameter tuning process. This involves conducting 16 random searches over the entire range of hyperparameters. We then use the validation set to identify the optimal hyperparameters for each algorithm. With these hyperparameters fixed, we repeat the experiments three times with different random seeds and report the average results along with their standard

---

[11]We also experimented with $\ell 1$ regularization, but it yielded poorer performance.

| Dataset | difFOCI+ERM | difFOCI+DRO | ERM | DRO | JTT | Mixup | IRM |
|---|---|---|---|---|---|---|---|
| MultiNLI | $\mathbf{77.6 \pm 0.1}$ | $\mathbf{77.5 \pm 0.2}$ | $66.9 \pm 0.5$ | $77.0 \pm 0.1$ | $69.6 \pm 0.1$ | $69.5 \pm 0.4$ | $66.5 \pm 1.0$ |
| CivilComments | $66.32 \pm 0.2$ | $\mathbf{70.3 \pm 0.2}$ | $64.1 \pm 1.1$ | $\mathbf{70.2 \pm 0.8}$ | $64.0 \pm 1.1$ | $65.1 \pm 0.9$ | $63.2 \pm 0.5$ |
| CelebA | $\mathbf{89.32 \pm 0.4}$ | $89.8 \pm 0.9$ | $65.0 \pm 2.5$ | $\mathbf{88.8 \pm 0.6}$ | $70.3 \pm 0.5$ | $57.6 \pm 0.5$ | $63.1 \pm 1.7$ |
| NICO++ | $\mathbf{47.10 \pm 0.7}$ | $46.3 \pm 0.2$ | $39.3 \pm 2.0$ | $38.3 \pm 1.2$ | $40.0 \pm 0.0$ | $43.1 \pm 0.7$ | $40.0 \pm 0.0$ |
| MetaShift | $83.1 \pm 0.5$ | $\mathbf{91.7 \pm 0.2}$ | $80.9 \pm 0.3$ | $86.2 \pm 0.6$ | $82.6 \pm 0.6$ | $80.9 \pm 0.8$ | $84.0 \pm 0.4$ |

Table 12: Worst group accuracy for benchmark methods on various datasets performed in Section 5.2. We can see that on almost all datasets, diFFOCI performs competitively.

deviations. This approach ensures a fair comparison between algorithms, where each is evaluated with its best possible hyperparameters, allowing for a reliable assessment of their performance. We provide brief dataset information below and in Table 10.

**CelebA** (Liang & Zou, 2022): A binary classification image dataset comprising over 200,000 celebrity face images. The task is to predict hair color (blond vs. non-blond), with gender serving as a spurious correlation. We employ standard dataset splits from prior work (Idrissi et al., 2022) and note that the dataset is licensed under the Creative Commons Attribution 4.0 International license.

**MetaShift** (Liang & Zou, 2022): A dataset creation method leveraging the Visual Genome Project (Krishna et al., 2017). We use the pre-processed Cat vs. Dog dataset, where the goal is to distinguish between the two animals. The spurious attribute is the image background, with cats more likely to appear indoors and dogs outdoors. We utilize the "unmixed" version generated from the authors' codebase.

**CivilComments** (Borkan et al., 2019): A binary classification text dataset aiming to predict whether an internet comment contains toxic language. The spurious attribute is the presence of references to eight demographic identities. We adopt the standard splits provided by the WILDS benchmark (Koh et al., 2021).

**MultiNLI** (Williams et al., 2017): A text classification dataset with three classes, targeting natural language inference relationships between premises and hypotheses. The spurious attribute is the presence of negation in the text, which is highly correlated with the contradiction label. We use standard train/val/test splits from prior work (Idrissi et al., 2022).

**NICO++** (Zhang et al., 2023): a large-scale dataset for domain generalization. Specifically, we focus on Track 1, which involves common context generalization. Our analysis is based on the training dataset, comprising 60 classes and 6 shared attributes: autumn, dim, grass, outdoor, rock, and water. To adapt this dataset for attribute generalization, we identify all pairs of attributes and labels with fewer than 75 samples and exclude them from our training data, reserving them for validation and testing purposes. For each attribute-label pair, we allocate 25 samples for validation and 50 samples for testing, while using the remaining data as training examples.

**CheXpert** (Irvin et al., 2019): a collection of chest X-ray images from Stanford University Medical Center, consisting of over 200,000 images. In this study, we use "No Finding" as the label, where a positive label indicates that the patient does not have any illness. Following previous research (Seyyed-Kalantari et al., 2021), we use the intersection of race (White, Black, Other) and gender as attributes. The dataset is randomly divided into 85

| Dataset | Method | Batch size | l.r. | N. epochs | w.d. | Val. loss | Test loss |
|---|---|---|---|---|---|---|---|
| Synth. Dataset | *vec*-(**dF1**) | full | 5e−2 | 2000 | 1e−1 | 0.02 ± 0.01 | 0.02 ± 0.02 |
| Toy Ex. 1 | | | 5e−3 | 1000 | 1e−4 | 0.01 ± 0.00 | 0.02 ± 0.00 |
| Toy Ex. 2 | *vec*-(**dF1**) | full | 10e−1 | 1000 | 1e−4 | 0.24 ± 0.00 | 0.25 ± 0.00 |
| Toy Ex. 3 | | | 5e−2 | 1000 | 5e−4 | 0.23 ± 0.00 | 0.24 ± 0.00 |
| Toy Ex. 1 | | | 5e−3 | 1000 | 1e−2 | 0.08 ± 0.01 | 0.08 ± 0.01 |
| Toy Ex. 2 | *NN*-(**dF1**) | full | 5e−3 | 1000 | 1e−2 | 0.02 ± 0.01 | 0.02 ± 0.01 |
| Toy Ex. 3 | | | 5e−4 | 1000 | 1e−2 | 0.18 ± 0.00 | 0.18 ± 0.01 |
| Spambase | | 2048 | 1e−2 | 100 | 1e−5 | 2.24 ± 0.14 | 2.56 ± 0.13 |
| Toxicity | | 4096 | 5e−2 | 50 | 1e−1 | 9.23 ± 1.96 | 11.61 ± 0.8 |
| QSAR | *vec*-(**dF1**) | 512 | 1e−2 | 10 | 5e−2 | 2.16 ± 0.14 | 2.54 ± 0.07 |
| Breast Canc. | | 2048 | 1e−3 | 50 | 1e−4 | 0.00 ± 0.00 | 0.00 ± 0.00 |
| Biblical | | 8 | 5e−4 | 50 | 1e−4 | 0.36 ± 0.02 | 0.48 ± 0.03 |
| Spambase | | 64 | 1e−4 | 10 | 1e−2 | 8.65 ± 0.91 | 9.61 ± 1.50 |
| Toxicity | | 512 | 5e−4 | 50 | 5e−2 | 1.97 ± 0.13 | 2.11 ± 0.11 |
| QSAR | *NN*-(**dF1**) | 512 | 1e−2 | 50 | 1e−1 | 2.52 ± 0.16 | 2.57 ± 0.19 |
| Breast Canc. | | 2048 | 5e−3 | 0 | 1e−6 | 0.34 ± 0.32 | 0.00 ± 0.00 |
| Biblical | | 128 | 1e−4 | 20 | 5e−4 | 0.69 ± 0.17 | 0.56 ± 0.04 |
| Student | *vec*-(**dF1**) | 64 | 1e−1 | 100 | 5e−4 | 8.03 ± 1.07 | 8.41 ± 0.82 |
| | *vec*-(**dF3**) | 256 | 5e−2 | 50 | 5e−3 | 7.65 ± 0.56 | 8.52 ± 0.89 |
| Bank | *vec*-(**dF1**) | 2048 | 5e−3 | 50 | 1e−5 | 2.61 ± 0.02 | 2.68 ± 0.04 |
| | *vec*-(**dF3**) | 256 | 5e−3 | 50 | 5e−4 | 2.59 ± 0.06 | 2.90 ± 0.07 |
| ACS Empl. | *vec*-(**dF1**) | 64 | 5e−2 | 50 | 5e−3 | 7.65 ± 0.08 | 7.81 ± 0.03 |
| | *vec*-(**dF3**) | 256 | 5e−3 | 50 | 1e−5 | 7.81 ± 0.01 | 7.97 ± 0.02 |
| ACS Inc. | *vec*-(**dF1**) | 1024 | 1e−2 | 10 | 1e−4 | 7.65 ± 0.01 | 7.65 ± 0.01 |
| | *vec*-(**dF3**) | 256 | 5e−2 | 100 | 5e−3 | 7.90 ± 0.01 | 7.92 ± 0.01 |

Table 13: Hyperparameter configurations used throughout the experiments.

| Method | Batch size | l.r. | reg. $\lambda$ | w.d. | Val. Acc. | Test Acc. | Val. WGA | Test WGA |
|--------|-----------|------|----------------|------|-----------|-----------|----------|----------|
| ERM | 8 | 1e−5 | 1e−3 | 5e−2 | 91.2 | 93.7 | 84.2 | 85.7 |
| gDRO | 32 | 1e−5 | 1e−1 | 1e−5 | 92.1 | 93.5 | 85.7 | 87.2 |

Table 14: Hyperparameter configurations for Waterbirds experiment with *NN*-(**dF2**) method.

## H ALGORITHMIC EXAMPLES

In this section, we give three concrete examples of Alg. 2 used in Sections 5.1-5.3 for completeness: using the *vec*-(**dF1**), *NN*-(**dF2**) and *vec*-(**dF3**) versions respectively.

---

**Algorithm 3** difFOCI: version *vec*-(**dF1**)

---

**Input:** Standardized input $\mathbf{X} \in \mathbb{R}^{n,p}$, $Y \in \mathbb{R}^n$
**Input:** learning rate $\gamma$, weight decay parameter $\lambda$, batch size $b$, cutoff parameter $\upsilon$, softmax parameter $\beta$
init. $\theta \sim \mathrm{N}(1, \sigma^2\, \mathrm{I}_p)$, with $\sigma^2 = 0.1$
**for** $t = 1, ..., n_{\text{iter}}$ **do**
    $\mathcal{L} \leftarrow -T_{n,\beta}(Y, \theta_t \odot \mathbf{X})$                    // Differentiable objective
    $\theta_{t+1} \leftarrow \theta_t - \gamma \mathrm{Adam}_{\mathrm{WD}_{\lambda,b}}(\mathcal{L})$              // Parameter update
$\theta_{\text{final}} = c(\theta_{n_{\text{iter}}}, \upsilon)$                         // Parameter clipping
**Output:** parameter $\theta_{\text{final}}$

---

Alg. 3 is version of difFOCI used in Section 5.1 for feature learning and domain shift experiment.

---

**Algorithm 4** difFOCI: version *NN*-(**dF2**)

---

**Input:** Standardized input $\mathbf{X} \in \mathbb{R}^{n,p}$, $Y \in \mathbb{R}^n$, group attribute $\mathbf{X}_G$
**Input:** learning rate $\gamma$, weight decay parameter $\lambda$, regularization strength $\eta$, batch size $b$, softmax parameter $\beta$, neural network $f_\theta(\cdot) = f_{LL_\theta}(f_{FE_\theta}(\cdot))$, where $f_{LL_\theta}$ and $f_{FE_\theta}$ denote the last layer and the feature extractor respectively, BCEWithLogits loss $\ell(\cdot, \cdot)$
init. NN parameters $\theta$
**for** $t = 1, ..., n_{\text{iter}}$ **do**
    $\mathcal{L}_1 \leftarrow \ell(Y, f_{LL_{\theta_t}}(f_{FE_{\theta_t}}(\mathbf{X})))$           // Standard BCEWithLogits Loss
    $\mathcal{L}_2 \leftarrow T_{n,\beta}(\mathbf{X}_G, f_{FE_{\theta_t}}(\mathbf{X}))$               // difFOCI regularizer
    $\mathcal{L} \leftarrow \mathcal{L}_1 + \eta \mathcal{L}_2$                    // Total loss calculation
    $\mathcal{L}^* \leftarrow w_{\text{gDRO}}(\mathcal{L})$ or $w_{\text{ERM}}(\mathcal{L})$     // Reweighting (in case of DRO)
    $\theta_{t+1} \leftarrow \theta_t - \gamma \mathrm{SGD}_{\mathrm{WD}_{\lambda,b}}(\mathcal{L}^*)$         // Parameter update
$\theta_{\text{final}} = c(\theta_{n_{\text{iter}}}, 0.1)$                   // Final parameter clipping
**Output:** neural network parameters $\theta_{\text{final}}$

---

Alg. 4 is version of difFOCI used in Section 5.2 for feature learning and domain shift experiment.

Alg. 5 is version of difFOCI used in the fairness Section 5.3.

## I FAIRNESS EXPERIMENTS

### I.1 EXPERIMENT IN SECTION 5.3

Here, we provide experimental details regarding the experiment in Section 5.3. In this study, we employed a data splitting approach where the dataset was divided into training, validation, and testing sets in a ratio of 75For our first network, we implemented a multi-layer perceptron (MLP) with three hidden layers, each featuring ReLU activations. We employed the BCEWithLogits loss function from PyTorch, along with the Adam optimizer as our optimization algorithm. The learning rate and weight decay were set as hyperparameters for the Adam optimizer. To predict sensitive attributes, we leveraged the last layer of the MLP and trained an additional three-layer MLP on top of it, again utilizing ReLU activations and the BCEWithLogits loss function from PyTorch. The Adam optimizer

---

**Algorithm 5** difFOCI: version *vec*-**(dF3)**

---

**Input:** Standardized input $\mathbf{X} \in \mathbb{R}^{n,p}$, $Y \in \mathbb{R}^n$, sensitive attribute(s) $\mathbf{X}_S$
**Input:** learning rate $\gamma$, weight decay parameter $\lambda$, batch size $b$, softmax parameter $\beta$
init. $\theta \sim \mathrm{N}(1, \sigma^2\,\mathrm{I}_p)$, with $\sigma^2 = 0.1$
**for** $t = 1, ..., n_{\mathrm{iter}}$ **do**
$\quad \mathcal{L} \leftarrow -T_{n,\beta}(Y, \theta_t \odot \mathbf{X} \mid \mathbf{X}_S)$ $\qquad\qquad\qquad$ // NN-based differentiable objective
$\quad \theta_{t+1} \leftarrow \theta_t - \gamma\mathrm{Adam}_{\mathrm{WD}_{\lambda,b}}(\mathcal{L})$ $\qquad\qquad\qquad\qquad$ // Parameter update
$\theta_{\mathrm{final}} = c(\theta_{n_{\mathrm{iter}}}, 0.1)$ $\qquad\qquad\qquad\qquad\qquad$ // Final parameter clipping
**Output:** parameter $\theta_{\mathrm{final}}$

---

|  | GUS | S.Per. | FPR | FDR | FWE | K.B. | UMAP | LDA | PCA | FOCI | *vec*-**(dF1)** | *vec*-**(dF3)** |
|---|---|---|---|---|---|---|---|---|---|---|---|---|
| Student | 13.30 | 14.14 | 11.36 | 11.64 | 11.36 | 10.53 | 8.87 | **8.31** | 8.59 | 10.25 | **8.41 ± 0.82** | 8.52 ± 0.89 |
| Bank | 3.45 | 3.19 | 3.19 | 3.19 | 3.19 | 3.10 | 3.32 | 3.01 | 3.32 | 3.01 | **2.68 ± 0.04** | 2.90 ± 0.07 |
| Income | 10.39 | 10.39 | 7.62 | 7.62 | 7.62 | 7.71 | **7.49** | 7.86 | 8.02 | 7.96 | 7.65 ± 0.01 | 7.92 ± 0.01 |
| Employment. | 12.09 | 11.23 | 8.31 | 8.31 | 8.31 | 8.41 | 14.60 | 8.67 | 9.01 | 8.43 | **7.81 ± 0.03** | 7.97 ± 0.02 |

Table 15: Fairness experiments and test log-loss. *vec*-**(dF1)** achieves best performance by not conditioning on sensitive attributes, though *vec*-**(dF3)** remains competitive even though it conditioning out the information regarding the sensitive data $\mathbf{X_S}$.

was used with learning rate and weight decay parameters, and the hidden dimensions had sizes of 128. When training using the (dF3)objective of diffoci, we employed a three-hidden-layer neural network, where all layers were of size 128. Throughout all experiments, the beta parameter was consistently set to 5.

## I.2 ANOTHER STUDY ON FAIRNESS

In this section, we perform similar experiments to Section 5.1, however we use the *vec*-**(dF3)** rather than **(dF1)**. We note that here we just experiment whether, by conditioning on $X_s$ we can still achieve good performance, which is affirmatively confirmed.

**Environments.** As in Section 5.3, we use Student dataset (Cortez & Silva, 2008), Bank Marketing dataset (Moro et al., 2014); and two ACS datasets (Ding et al., 2021), *(iii)* Employment and *(iv)* Income. Again, the performance is measured using Logistic Regression (Cox, 1958).

**Despite conditioning out sensitive information, difFOCI delivers solid performance.** From Table 15, we see that *vec*-**(dF3)** demonstrates strong performance, regardless of whether we condition on the sensitive data or not. Both algorithms outperform other methods, and, expectedly, we observe a slight decrease in performance when conditioning on the sensitive attribute(s). For the Student dataset, conditioning on sex leads to the exclusion of seven additional features (a total of 11 out of 30), while for the Bank Marketing dataset, conditioning on marital status results in the exclusion of one additional feature (a total of 1 out of 20).

**difFOCI might be useful in intersectional fairness.** In both ACS datasets, conditioning on both sensitives led to the exclusion of previously included features (when conditioning on just one sensitive), as shown in Table 9, in Appendix E. This reveals that the additional features, excluded only after considering both sensitives, might contain intertwined relationships with the two sensitives, providing an interesting avenue for intersectional fairness research (Gohar & Cheng, 2023). We leave this as future work.

## J CHOICE FOR THE REGULARIZATION PARAMETER

In this section, we provide empirical evidence of for our parameter $\beta$ choice (we fixed it to 5), highlighting that although we might observe minor improvements by tuning the parameter, the performance is consistent.

We observe that difFOCI exhibits robust performance across a range of values for the hyperparameter $\beta$. As long as $\beta$ is set within a reasonable range, avoiding extreme values that either zero out gradients

| $\beta$: | $1e-5$ | $1e-3$ | $1.$ | $5$ | $100$ | $1e5$ | $1e7$ | Standard |
|---|---|---|---|---|---|---|---|---|
| (dF2) ERM Avg. Acc. | $97.1\pm0.4$ | $97.0\pm0.7$ | $94.2\pm0.3$ | $93.7\pm0.1$ | $94.6\pm0.1$ | $97.4\pm0.1$ | $97.4\pm0.2$ | $97.3\pm0.2$ |
| WGA | $61.0\pm0.2$ | $62.3\pm0.8$ | $84.8\pm0.8$ | $85.7\pm0.8$ | $85.7\pm0.5$ | $63.9\pm0.8$ | $61.2\pm1.0$ | $60.0\pm0.5$ |
| (dF2) DRO Avg. Acc. | $97.2\pm0.1$ | $97.5\pm0.7$ | $93.9\pm0.2$ | $93.5\pm0.5$ | $93.6\pm0.7$ | $97.5\pm0.3$ | $97.2\pm0.1$ | $97.4\pm0.4$ |
| WGA | $75.7\pm1.0$ | $97.2\pm0.3$ | $90.0\pm0.4$ | $87.2\pm0.6$ | $87.0\pm0.3$ | $77.1\pm0.6$ | $76.9\pm0.3$ | $76.9\pm0.8$ |

Table 16: Results for various $\beta$ on Waterbirds dataset. The results for reasonable values of $\beta$ yield similar performance, and very large or small values result the performance falling back to the standard ERM or DRO performance.

| $\beta$: | $1e-5$ | $1e-3$ | $1.$ | $5$ | $100$ | $1e5$ | $1e7$ | Standard |
|---|---|---|---|---|---|---|---|---|
| (dF2) ERM Avg. Acc. | $91.2\pm0.7$ | $91.5\pm0.9$ | $92.3\pm0.2$ | $92.1\pm0.2$ | $91.7\pm0.3$ | $91.4\pm0.2$ | $91.3\pm0.1$ | $91.3\pm0.5$ |
| WGA | $81.1\pm0.2$ | $81.2\pm0.1$ | $83.3\pm0.2$ | $83.1\pm0.5$ | $83.1\pm0.7$ | $80.6\pm0.1$ | $81.3\pm0.3$ | $80.9\pm0.3$ |
| (dF2) DRO Avg. Acc. | $88.8\pm0.2$ | $90.0\pm0.4$ | $91.9\pm0.3$ | $91.8\pm0.3$ | $91.8\pm0.1$ | $88.7\pm0.3$ | $88.9\pm0.2$ | $89.0\pm0.2$ |
| WGA | $86.1\pm0.3$ | $86.2\pm0.4$ | $91.5\pm0.3$ | $91.7\pm0.2$ | $91.9\pm0.3$ | $85.8\pm0.2$ | $85.9\pm0.6$ | $86.2\pm0.6$ |

Table 17: Results for various $\beta$ on MetaShift dataset. The results for reasonable values of $\beta$ yield similar performance, and very large or small values result the performance falling back to the standard ERM or DRO performance.

or result in a uniform distribution from the softmax function, difFOCI consistently delivers robust results. This is evident in Tables 17 and 16, which present results on the MetaShift and Waterbirds datasets, respectively. Our experiments show that tuning $\beta$ leads to only minor performance improvements, which are largely statistically insignificant. Furthermore, setting $\beta$ to extreme values causes the estimator $T(X_G, f_\theta(X))$ to degenerate to a constant, effectively reducing difFOCI to standard ERM performance.

| Method | Train Accuracy | Test Accuracy | OOD Accuracy | Difference |
|---|---|---|---|---|
| Standard | $85.20 \pm 2.0$ | $82.75 \pm 1.9$ | $70.2 \pm 1.6$ | 12.55 |
| difFOCI with 75% feats. | $82.66 \pm 1.2$ | $81.7 \pm 2.7$ | $68.95 \pm 0.8$ | 12.22 |
| difFOCI with 50% feats. | $80.19 \pm 2.4$ | $79.4 \pm 1.0$ | $67.9 \pm 1.2$ | 11.5 |
| difFOCI with 25% feats. | $79.55 \pm 2.1$ | $78.72 \pm 1.1$ | $65.40 \pm 2.8$ | 13.32 |

Table 18: Difference between standard predictive accuracy using ResNet-50 on CIFAR10 and CIFAR10.1

Table 19: Difference between standard predictive accuracy using Resnet-50 on various DomainNet datasets

| Dataset | Method | Train Accuracy | Test Accuracy | OOD Accuracy | Difference |
|---|---|---|---|---|---|
| Real vs. Sketch | Standard | $88.72 \pm 1.7$ | $78.93 \pm 0.6$ | $29.34 \pm 2.1$ | 49.59 |
| | difFOCI, clip at 0.1 | $85.58 \pm 1.3$ | $77.50 \pm 1.3$ | $27.58 \pm 1.5$ | 49.92 |
| Clipart vs. Sketch | Standard | $89.98 \pm 2.1$ | $61.85 \pm 0.6$ | $39.43 \pm 2.1$ | 22.42 |
| | difFOCI, clip at 0.1 | $88.95 \pm 1.6$ | $61.96 \pm 1.3$ | $40.34 \pm 1.8$ | 21.92 |
| Sketch vs. Quickdraw | Standard | $66.51 \pm 1.0$ | $53.17 \pm 0.9$ | $7.01 \pm 1.9$ | 46.16 |
| | difFOCI, clip at .1 | $65.54 \pm 1.7$ | $52.97 \pm 0.8$ | $6.98 \pm 1.2$ | 45.99 |

## K  DIFFOCI'S ROBUSTNESS TO DOMAIN SHIFT

This section presents experimental results on feature selection using difFOCI's objective **(dF1)** on CIFAR10/10.1 (Recht et al., 2018) and DomainNet datasets (Peng et al., 2019), specifically examining the Real vs Sketch, Clipart vs Sketch, and Sketch vs Quickdraw domain shifts. The results, summarized in Tables 18-19 (for CIFAR10 vs CIFAR10.1 and DomainNet respectively) demonstrate that difFOCI maintains consistent performance across distribution shifts, with the selected features exhibiting similar performance differences as the full dataset. This consistency highlights difFOCI's ability to effectively handle distribution shifts.

