# OpenReview forum: "A Differentiable Rank-Based Objective for Better Feature Learning"
_ICLR.cc/2025/Conference — ICLR 2025 Poster_

### Official Review · Reviewer_P7Dz · 2024-10-30

**Soundness:** 2
**Presentation:** 3
**Contribution:** 2
**Rating:** 6
**Confidence:** 2

**Summary:**

This paper introduces Chatterjee’s coefficient into feature learning by proposing a differentiable formulation to estimate it. The conditional dependence metric proposed by the authors is applicable to feature learning and enhances performance in downstream tasks.

**Strengths:**

1. The proposed correlation estimation term is highly versatile and “plug-and-play,” making it adaptable for feature selection, predictor training, and as a regularizer to mitigate spurious correlations.
2. The paper is well-organized, with clear sections on methodology, relevant theorems, and extensive evaluation experiments that strengthen the contribution.

**Weaknesses:**

1. There is a disconnect between the theoretical foundation and its practical application, leading to a potentially vacuous theorem. The assumption that the coefficient \(\beta\) of the Softmax function is infinite is impractical, as it is only set to values like 5 in the experiments.
2. Symbol definitions are ambiguous, as discussed in the questions below.
3. The comparison of the proposed method focuses on conventional methods such as LDA and PCA. It would be beneficial to include comparisons with more state-of-the-art methods.

**Questions:**

1. The definition of the rank \( r \) in Line 096 is unclear. What is meant by \( Y_{(j)} \leq Y_{(i)} \)?
2. Line 076 declares \(\sigma\) as a Sigmoid function, yet it denotes the Softmax function. Could the authors clarify this discrepancy?

---

> ### Author Response · Authors · 2024-11-21
>
> Thank you for carefully reviewing our paper. In the following, we aim to address your concerns.
>
>
>
> > There is a disconnect between the theoretical foundation and its practical application, leading to a potentially vacuous theorem. The assumption that the coefficient (\beta) of the Softmax function is infinite is impractical, as it is only set to values like 5 in the experiments.
>
>
> Your feedback highlights a crucial aspect of difFOCI. Our theorem establishes that in the infinite $\beta$ regime, we converge to the estimator from [1], which in the infinite $n$ limit recovers the original measure of conditional dependence suggested by [2]. However, and we should have clarified this more carefully, there are two key points:
>
> 1) Theoretically, our estimator is more versatile as it provides a generalization of the estimator in [1].
> 2) In practice, we benefit from $\beta$ being finite as increasing it will result in softmax transitioning to a hardmax, causing the gradients to zero out. Therefore, in practice we want $\beta$ finite.
>
> The point that we initially intended concerns difFOCI’s robust performance to the selected value of $\beta$. Provided that it is set within a reasonable range - not too large to zero out gradients or too low as in this results in softmax yielding an uniform distribution - difFOCI demonstrated robust results across various values of this hyperparameter. We demonstrate this in Table 3 below on MetaShift dataset (you can find additional example in Appendix J): tuning $\beta$ yields minor performance gains, although these are mostly not statistically significant. Also, extreme $\beta$ values cause the estimator $T(X_G, f_\theta(X))$ to equal a constant (as mentioned before), which degrades it yielding standard ERM performance (these are marked in italic in the Table 3. below):
>
>
>
> ||Beta:|1e-5|1e-3|1.|5|10|100|1e5|1e7|Standard
> |---------|-----|-----|---|---|---|----|----|----|----|---|
> |(dF2) ERM|Avg. Acc.|*91.2±0.7*|*91.5±0.9*|92.3±0.2|92.1±0.2|92.0±0.4|91.7±0.3|*91.4±0.2*|*91.3±0.1*|91.3±0.5|
> |(dF2) ERM|WGA|*81.1±0.2*|*81.2±0.1*|83.3±0.2|83.1±0.5|83.0±0.5|83.1±0.7|*80.6±0.1*|*81.3±0.3*|80.9±0.3|
> |(dF2) DRO|Avg. Acc.|*88.8±0.2*|*90.0±0.4*|91.9±0.3|91.8±0.3|92.0±0.4|91.8±0.1|*88.7±0.3*|*88.9±0.2*|89.0±0.2|
> |(dF2) DRO|WGA|*86.1±0.3*|*86.2±0.4*|91.5±0.3|91.7±0.2|91.8±0.2|91.9±0.3|*85.8±0.2*|*85.9±0.6*|86.2±0.6|
> **Table 3. Results for various $\beta$ on MetaShift**
>
> We are interested in knowing if you find the results convincing or believe additional explanation is necessary. We appreciate bringing this to our attention as further clarification was indeed needed.
>
> > Symbol definitions are ambiguous, as discussed in the questions below.
>
> We address these below - thank you for detailed attention to our paper!
>
> > The definition of the rank ( r ) in Line 096 is unclear. What is meant by ( Y_{(j)} \leq Y_{(i)} )?
>
> Consider a dataset where the $X_i$ and $Y_i$ values have no ties. Arrange the data pairs $(X_1, Y_1), (X_2, Y_2), \ldots, (X_n, Y_n)$ such that the $X$ values are in ascending order: $X_{(1)} \leq X_{(2)} \leq \ldots \leq X_{(n)}$. This yields $(X_{(1)},Y_{(1)}),..., (X_{(n)},Y_{(n)})$.
> Now, since there are no ties among the $X_i$ values, this ordering is unique. Therefore, we define $r_i$ as the rank of $Y_{(i)}$, the count of $j$ such that $Y_{(j)} \leq Y_{(i)}$. In other words, $r_i$ represents the position of $Y_{(i)}$ when $Y_i$’s are sorted in ascending order. Thank you for raising this - we have also added this clarification to the manuscript (as well as the case in which there are ties in the data).
>
> > Line 076 declares (\sigma) as a Sigmoid function, yet it denotes the Softmax function. Could the authors clarify this discrepancy?
>
> We apologize - we meant the Softmax function and have amended this accordingly.
>
> > The comparison of the proposed method focuses on conventional methods such as LDA and PCA. It would be beneficial to include comparisons with more state-of-the-art methods.
>
>
> We appreciate your feedback: accordingly, we have conducted 5 additional experiments. As shown in Table 1 (in the main response), we performed experiments on both text and image datasets of varying sizes, architectures and included additional three baseline comparisons with state-of-the-art methods: Just Train Twice, Mixup and Invariant Risk Minimization. The experiments illustrate difFOCI’s compatibility with contemporary architectures, such as ViT-B or BERT, which allow difFOCI to achieve competitive performance.
>
> We hope that our answers and newly conducted experiments addressed your concerns. Please let us know if you have any remaining questions.
>
> ---
>
> [1] Azadkia, M. and Chatterjee, S., 2021. A simple measure of conditional dependence. The Annals of Statistics, 49(6)
>
> [2] Dette, H., Siburg, K.F. and Stoimenov, P.A., 2013. A Copula‐Based Non‐parametric Measure of Regression Dependence. Scandinavian Journal of Statistics

---

> > ### Comment · Reviewer_P7Dz · 2024-11-27
> >
> > Thanks for the detailed response that resolves all concerns. I am inclined to increase my score to 6.

---

### Official Review · Reviewer_uzzo · 2024-11-03

**Soundness:** 3
**Presentation:** 2
**Contribution:** 2
**Rating:** 6
**Confidence:** 3

**Summary:**

The paper introduces an algorithm (difFOCI) that gives a measure of correlation based on Feature Ordering by Conditional Independence (FOCI). The method is differentiable and can be used in ML methods. It is used in 3 ways:

dF1) maximize T(Y,fθ(X));

dF2) minimize loss + T(X_G,fθ(X)) such that the features are independent of group features

dF3) maximize T(Y,fθ(X)|X_S) such that features are correlated with Y when conditioned on X_s

Synthetic and more real data experiments are performed for all cases.

**Strengths:**

* S1. The proposed method is sound and has good potential for feature selection and feature debiasing

* S2. The method has mainly good results in synthetic and some real datasets.

**Weaknesses:**

* W1. The experiments on real datasets are not that strong.

* W1.1 For the spurious correlation experiments, Waterbirds dataset is a small and simple dataset and a successful method should be tested on other datasets besides it. How many seeds were used? Was the same protocol used to select hyperparameters for the baselines and the proposed method? The benchmark used in [A] can be used to evaluate the proposed method more rigorously. The results of the method will be more reliable if multiple datasets from [A] were used, together with similar hyperparameter selection and evaluation.


* W2. For the fairness experiments, some additional explanations are needed.

* W2.1 Are the explicit features Xs a subset of X?

* W2.2 The authors state that “the predictor should be independent of one or more sensitive features”. How is this enforced? By maximizing T (Y , θ ⊙ X | Xs ) the model would learn to predict the target when we know Xs. Is this right? Can the authors explain this?

* W2.3 Is there a measure to evaluate if the model relies on sensitive features? At the moment the results only report performance, and it is not clear if these results incorporate sensitive features or not.



Minor:
* In the context of Waterbirds, group refers to the combination of class and spurious variable (background). So X_G should be denoted as a spurious variable, not a group variable.


[A] Yang et al. "Change is hard: A closer look at subpopulation shift." (2023)

**Questions:**

See weak points.

---

> ### Author Response · Authors · 2024-11-21
>
> We sincerely appreciate the time and effort you dedicated to reviewing our paper. We were encouraged by your recognition of difFOCI's potential. In response to your constructive feedback, we undertook a thorough revision of the message some sections convey and conducted several further experiments.
>
> > Waterbirds dataset is a small and simple dataset and a successful method should be tested on other datasets besides it. How many seeds were used? Was the same protocol used to select hyperparameters for the baselines and the proposed method? The benchmark used in [A] can be used to evaluate the proposed method more rigorously. The results of the method will be more reliable if multiple datasets from [A] were used, together with similar hyperparameter selection and evaluation.
>
> This is a very valid critique - thank you for pointing us to the relevant paper. In response, we incorporated all datasets from [A] (except ImageNetBG and Living17 as they do not contain attribute data). We are still waiting for access to medical data in [A] (as the datasets are not publicly available) and will include them once we obtain access.
>
> Our findings are presented in Table 1 (in main response), with experimental procedures outlined in Appendix G (together with the Waterbirds hyperparameter configuration). To ensure consistency and comparability, we exactly replicated experimental configuration used in [A], including random search over same hyperparameter distribution, same train-validation-test split, and reporting averages with std.s across 3 random seeds. We would like to express our gratitude to the reviewer for recommending this paper, as their extensive codebase (with various architectures, datasets and algorithms) helped us in obtaining meaningful results in a timely manner. We also added this acknowledgement to the paper.
>
> >  W2. For the fairness experiments, some additional explanations are needed.
>
> >  W2.1 Are the explicit features Xs a subset of X?
>
>
> The data is as follows $(X, X_s, y)$, and therefore $X$ does not contain $X_s$ . For example, in ASCI dataset $X=\\{\text{occupation, working class, place of birth, etc.}\\}$, $y=$income and $X_s=\\{\text{race, sex}\\}$. When we use difFOCI (dF3) to optimize $T(y, f_\theta(X) | X_s)$, where $X_s$ is only used in the conditioning part. We have clarified this in the paper.
>
> >  W2.2 The authors state that “the predictor should be independent of one or more sensitive features”. How is this enforced? By maximizing T (Y , θ ⊙ X | Xs ) the model would learn to predict the target when we know Xs. Is this right? Can the authors explain this?
>
> >  W2.3 Is there a measure to evaluate if the model relies on sensitive features? At the moment the results only report performance, and it is not clear if these results incorporate sensitive features or not.
>
> These are valid points: upon reflection, our assertion about difFOCI achieving statistical independence was an overstatement. We have removed any mention of statistical independence, toning down the assertion. This was an oversight on our part - we apologize for this. Our original motivation for $T(y, f_\theta(X)|X_s)$ was empirical, as preliminary experiments demonstrated promising performance.
>
> Specifically, we trained two NNs, first on $(X,y)$ without $X_s$ and second using difFOCI with $T(y, f_\theta(X)|X_s)$. We used the final layers of the two NNs to predict $X_s$ and, as can be seen in Table 2 (in main response), difFOCI significantly reduces predictability of $X_s$, sometimes to chance level, without significantly impacting accuracy on y (sometimes even slightly improving it). We have included this in the manuscript.
> Finally, we would like to point out that difFOCI’s primary contribution lies in its regularizer form (dF2). We clarified this in the manuscript, noting (for the fairness Section 5.3): *“This section, while not the primary focus of our contribution, offers a complementary illustration of the difFOCI objective's versatility through a heuristic example. We found that this form (dF3) preserves the performance of the chosen parameterization while significantly reducing its predictivity of the sensitive attribute.”*
>
> >  In the context of Waterbirds, group refers to the combination of class and spurious variable (background). So X_G should be denoted as a spurious variable, not a group variable.
>
> Thank you for pointing this out, we corrected the manuscript.
>
> Once again, we would like to thank you for your constructive feedback - we believe that addressing your critiques helped us significantly strengthen the contribution of difFOCI’s.

---

> > ### Comment · Reviewer_uzzo · 2024-11-26
> > **Post rebuttal**
> >
> > I thank the authors for their rebuttal. I appreciate the additional experiments and explanations.
> >
> > Regarding (dF3), the revisions still says "is a conditioning objective, allowing to learn features that contain information about the response **only** after conditioning out the sensitive information XS". The "only" part is not clear. As far as I understand, we don't have an explicit constraint that enforces the features to not be predictive of Xs. Experimentally it seems like the learned features are less predictive of Xs. It should be more clear that the method does not enforce "conditioning out sensitive information", but offers an optimization that seem to favour solutions that are less predictive of Xs.
> >
> > I will increase my score to 6.

---

> > > ### Author Response · Authors · 2024-11-27
> > >
> > > Thank you for your response.
> > >
> > > You raise a valid point. We have replaced this sentence as well with: *Using NN-dF3, we optimize $T_{n,\beta}(Y,  f_\theta(\mathbf{X}) \mid \mathbf{X}_s)$ to learn features that are informative about $Y$, offering an optimization that heuristically seems to favor solutions less predictive of $\mathbf{X}_s$.*
> > >
> > > We once again thank you for your careful consideration.

---

### Official Review · Reviewer_cLoX · 2024-11-04

**Soundness:** 3
**Presentation:** 3
**Contribution:** 2
**Rating:** 6
**Confidence:** 3

**Summary:**

This paper uses existing statistical methods to better understand feature learning from data, by modifying an existing model-free variable selection method to a trainable version. Experiments on toy examples and real world datasets show the effectiveness of the proposed method.

**Strengths:**

1. The motivation of the paper is well stated.
2. The paper is well-structured and well-written.
3. Providing results on both toy experiments and real world datasets makes the paper more solid.

**Weaknesses:**

1. The real world datasets seem to be out-dated, where the latest one was released in 2019. It would be more convincing to presents results on more recent are more complex datasets, such as those in WILDS benchmark.

2. This paper only considers one model architecture, i.e., ResNet-50. With the increasing usage of Transformer-based models, it is also important to show the effectiveness on more complex models.

3. Simply showing the improved performance on worst group accuracy does not sufficiently suggest the decrease reliance on the spurious features. Note that, the overall performance of the proposed method shows consistently lower performance \(2 - 4%\) for average accuracy. It would be suggested to conduct the synthetic experiments in [1].

4. Another concern is that the experiments are conducted on the case where the train test set share the same distribution. Given that the feature selector is a trained NN, it is interesting to show if such a method maintains its performance of feature selection where the train test set have a different distribution.  For example, CIFAR10 vs CIFAR10.1, DomainNet-real vs DomainNet-Sketch? as in [2].

I would be happy to increase my score if some of my concerns are resolved.

[1] The Pitfalls of Simplicity Bias in Neural Networks, NeurIPS, 2020.

[2] A CLOSER LOOK AT MODEL ADAPTATION USING FEA- TURE DISTORTION AND SIMPLICITY BIAS, ICLR, 2023.

**Questions:**

See weakness.

---

> ### Author Response · Authors · 2024-11-21
>
> We thank you for your thorough feedback. In your review, you raised several valid points.
>
> >  1. The real world datasets seem to be out-dated, where the latest one was released in 2019. It would be more convincing to presents results on more recent are more complex datasets, such as those in WILDS benchmark.
>
>
> This is a valid concern - we acknowledged this and expanded our study to five new datasets, out of which two date to 2022 and another one includes a WILDS dataset.
>
> >  2. This paper only considers one model architecture, i.e., ResNet-50. With the increasing usage of Transformer-based models, it is also important to show the effectiveness on more complex models.
>
>
> The newly added experiments encompass multiple modalities (images and text) and incorporate transformer architectures: ViT-B pretrained with DINO or CLIP and BERT pretrained on English corpus and Wikipedia. The datasets contain up to 300k samples and up to 60 classes. As can be seen from Table 1 (in the main response), difFOCI exhibits robust performance across all datasets.
>
> >  3. Simply showing the improved performance on worst group accuracy does not sufficiently suggest the decrease reliance on the spurious features. Note that, the overall performance of the proposed method shows consistently lower performance (2 - 4%) for average accuracy. It would be suggested to conduct the synthetic experiments in [1].
>
> This is an important point we would like to address. With the new datasets, we found this decrease in average accuracy to be dataset-dependent; the new experiments reveal an improvement in average accuracy (as well as worst-group) on four out of five datasets (Table 1).
>
> The provided reference is an interesting study of simplicity bias through concatenation of MNIST and CIFAR datasets. While difFOCI does not directly extend to this due to the absence of sensitive/group attributes, your observation on solely reporting improved worst-group accuracy remains valid. We hope to alleviate this concern with the additional experiments presented in Table 2 (in the main response). Here, we trained two NNs to predict $y$: one on the whole dataset $(X,y)$ but without sensitive attributes $X_s$ and one on features obtained through difFOCI (dF3) - the experiment is detailed in Appendix G. Afterwards, we use the last layers of these two NNs to predict the sensitive attribute $X_s$. From Table 2, we see that the network trained on features obtained with (dF3) has a significant reduction in predictability of the sensitive $X_s$, sometimes down to chance level. Interestingly, training on difFOCI’s features does not compromise overall task performance on $y$ (in fact, sometimes we even observe a slight gain in accuracy).
>
>
> > 4. Another concern is that the experiments are conducted on the case where the train test set share the same distribution. Given that the feature selector is a trained NN, it is interesting to show if such a method maintains its performance of feature selection where the train test set have a different distribution. For example, CIFAR10 vs CIFAR10.1, DomainNet-real vs DomainNet-Sketch? as in [2].
>
> This is an interesting angle that we have not previously considered. We thank the reviewer for the suggestion: this motivated our decision of including NICO++ dataset in our evaluation, which is, as its authors state, ‘specifically designed to facilitate out-of-distribution (OOD) generalization in visual recognition’ [1] and is often characterized as a distribution-shift dataset (e.g., see [2,3]). In Appendix K, we also added experiments for feature selection on CIFAR10/10.1 and DomainNet (Real vs Sketch, Clipart vs Sketch and Sketch vs Quickdraw) showcasing difFOCI consistently maintains its performance. We hope that these prove difFOCI's effectiveness in handling distribution shifts.
>
> We thank the reviewer for the thoughtful suggestions. The four points raised helped us in strengthening our paper, and we hope that the reviewer will find the updated version to be a substantial improvement.
>
> ---
>
> [1] Zhang, X., He, Y., Xu, R., Yu, H., Shen, Z. and Cui, P., 2023. Nico++: Towards better benchmarking for domain generalization. In Proceedings of the IEEE/CVF conference on computer vision and pattern recognition (pp. 16036-16047).
>
> [2] Gulrajani, I. and Lopez-Paz, D., 2020. In search of lost domain generalization. International Conference on Learning Representations, 2021.
>
> [3] Zhang, X., He, Y., Xu, R., Yu, H., Shen, Z. and Cui, P., 2023. Nico++: Towards better benchmarking for domain generalization. In Proceedings of the IEEE/CVF conference on computer vision and pattern recognition (pp. 16036-16047).

---

> ### Comment · Reviewer_cLoX · 2024-11-27
>
> Dear authors,
>
> Thank you for your efforts. My concerns have been addressed by the added experiments. I will increase my score to 6.
>
> Best regards,
>
> Reviewer cLoX

---

### Official Review · Reviewer_ZQah · 2024-11-04

**Soundness:** 3
**Presentation:** 2
**Contribution:** 2
**Rating:** 6
**Confidence:** 3

**Summary:**

The paper introduces a new method called difFOCI, which bases on an existing approach called FOCI. FOCI is a tool for choosing important features from data based on their statistical relationships, but it is not easy to use in some deep learning tasks because it does not adapt well to different models.  Motived by this, difFOCI creates a flexible, learnable version of FOCI that can be used in more situations.

This submision uses difFOCI in three ways: as a tool to pick important features, as part of a model that learns using a neural network, and as a method to help models learn better by reducing irrelevant correlations in the data. To validate this, this submission tests difFOCI on a range of tasks, from simple examples to more complex problems like understanding which parts of images are important in neural networks. Also, difFOCI is shown to be helpful in making fairer predictions by letting models classify without depending on sensitive information.

**Strengths:**

- The motivation is clear to me. FOCI is a tool for selecting important features from data based on their statistical relationships. However, it is not differentiable, which hinders its use in deep neural networks. To address this limitation, this submission proposes a differentiable, parametric approximation.
- This submission provides a clear definition of difFOCI. The toy examples offer some intuition into how difFOCI works.
- The three applications are well-chosen. They effectively demonstrate the usefulness of difFOCI.

**Weaknesses:**

- While some real-world datasets are used in the experiments to demonstrate the effectiveness of difFOCI, it is unclear if it can be extended to large-scale datasets. Specifically, the datasets in Section 5.1 are small-scale, and the neural networks or learning algorithms used are relatively simple. The Waterbird task, for example, is simpler compared to multi-class tasks. Please discuss the scalability and generalization potential of difFOCI.
- The fairness study is interesting; however, the dataset and task here are relatively simple. Can difFOCI be applied in large-scale settings?
- The visualization in Figure 2 is not very clear. What is the main takeaway from this figure? Additional visualization examples would be helpful.

**Questions:**

- Please discuss the scalability of difFOCI on large-scale datasets and its applicability to more challenging tasks.
- Additionally, please clarify the purpose of the visualization in Figure 2.

---

> ### Author Response · Authors · 2024-11-21
>
> Thank you for the encouraging feedback, we are glad that you found the motivation clear and the applications well chosen.
>
> > Please discuss the scalability and generalization potential of difFOCI.
>
> We agree with your concerns regarding the scalability of our method. In response, we conducted further experiments on five datasets of varying sizes (up to 300k data samples), complexity (up to 60 classes) and modalities (text and image). In addition to ResNet-50, we incorporated attention-based architectures (e.g., BERT and ViT-B) utilizing various pretraining strategies (e.g., DINO and CLIP). We also added Just Train Twice, Mixup and Invariant Risk Minimization as additional baselines. As illustrated in Table 1 (in main response), difFOCI demonstrates competitive performance, both in terms of average accuracy and worst-group accuracy.
>
> > The fairness study is interesting; however, the dataset and task here are relatively simple. Can difFOCI be applied in large-scale settings?
>
> To address this, we chose CivilComments (~250k instances) as one of the added datasets. The task concerns prediction of toxic language, with sensitive attributes related to race, sexual orientation, or religion. The dataset is commonly used for fairness evaluations, e.g., in [1,2]. Across all approaches on CivilComments, difFOCI+ERM yields highest overall accuracy, while difFOCI+DRO results in best worst group accuracy. We hope that this provides further evidence of difFOCI’s potential to contemporary fairness applications.
>
> > The visualization in Figure 2 is not very clear. What is the main takeaway from this figure? Additional visualization examples would be helpful.
>
> The purpose of this figure is to provide a qualitative example, complementing quantitative analysis in Table 3, visually demonstrating that a difFOCI-trained model ignores spurious features (background), focusing on the relevant features (foreground). Without difFOCI, we can see from Figure 2 that models heavily rely on background, while with difFOCI this problem is effectively resolved. We’ve updated the caption of Figure 2 to summarize this key message. Note that this is not specific to this single example: in Appendix E1, we present 10 randomly-selected visualizations showing similarly strong effects.
>
> Thank you for your feedback. We made corresponding revisions to the manuscript, including the addition of scalability and fairness experiments. We would appreciate your assessment of whether these changes have adequately addressed your concerns, and welcome any further feedback or questions you may have.
>
> ---
>
> [1] Dige, O., Arneja, D., Yau, T.F., Zhang, Q., Bolandraftar, M., Zhu, X. and Khattak, F., 2024, November. Can Machine Unlearning Reduce Social Bias in Language Models?. In Proceedings of the 2024 Conference on Empirical Methods in Natural Language Processing: Industry Track (pp. 954-969).
>
> [2] Villate-Castillo, G., Lorente, J.D.S. and Urquijo, B.S., 2024. A systematic review of toxicity in large language models: Definitions, datasets, detectors, detoxification methods and challenges.

---

> > ### Comment · Reviewer_ZQah · 2024-11-27
> > **The Newly Added Evaluation Is Helpful**
> >
> > Thank you for the efforts in the rebuttal. I found the response helpful in addressing my initial concern regarding scalability. The newly added evaluations are well-executed and further demonstrate the effectiveness of the proposed Rank-Based Objective. I recommend incorporating these experiments into the revised version of the paper. I have updated my score to 6.

---

### Author Response · Authors · 2024-11-21
**Updates to the paper in response to the reviewers**

We would like to express our gratitude for the reviewers’ thorough evaluations and constructive feedback. While the reviewers found the motivation behind difFOCI clear, they collectively identified two key areas for improvement:
- **More comprehensive experiments**: all reviewers recommended including additional architectures and employing larger, more diverse datasets and further benchmarks.
- **Additional fairness evidence**: reviewers ZQah, cLoX, and uzzo requested additional evidence to demonstrate the capabilities of difFOCI in reducing bias/spurious correlations.

*(Edit 27/11: added medical CheXpert dataset.)*
In response to the first point, reviewers’ consensus prompted us to conduct further experiments. We evaluated difFOCI across five additional datasets, which vary in size, number of classes, modality, architecture, and pretraining displayed in Table 1a:

|Dataset|Year|Size|N. Classes|Modality|Architecture|
|--|--|--|---|--|--|
|MultiNLI|2017|300k|3|Text|BERT|
|CivilComments|2019|250k|2|Text|BERT|
|CelebA|2015|200k|2|Image|ResNet-50 w. ImageNet-1K|
|NICO++|2022|90k|60|Image|ViT-B w. DINO|
|MetaShift|2022|3.5k|2|Image|VIT-B w. CLIP|
|CheXpert|2019|225k|2|Image|ResNet-50 w. ImageNet-21K|
**Table 1a: Dataset Overview**

We further added three benchmarks: Just Train Twice (JTT) [1], Mixup [2] and Invariant Risk Minimization (IRM) [3]. Our results show that difFOCI (with ERM and DRO) achieves competitive performance, both in terms of average accuracy (Table 1b) and worst group accuracy (Table 1c). We appreciate the reviewers suggesting this enhancement.

|Average acc.|difFOCI+ERM|difFOCI+DRO|ERM|DRO|JTT|Mixup|IRM|
|--|---|--|--|--|--|--|--|
|MultiNLI|**81.98±0.2**|**81.82±0.5**|81.4±0.1|80.2±0.6|81.2±0.4|80.7±0.1|77.7±0.3|
|CivilComments|**86.28±0.1**|81.90±0.3|85.7±0.4|82.3±0.4|84.3±0.5|84.9±0.3|85.4±0.2|
|CelebA|94.41±1.1|92.9±2.1|94.9±0.2|93.1±0.6|92.4±1.6|**95.7±0.2**|94.5±1.0|
|NICO++|**85.7±0.3**|**85.8±0.5**|84.7±0.6|83.0±0.1|**85.3±0.1**|84.2±0.4|84.7±0.5|
|MetaShift|**92.1±0.2**|**91.8±0.3**|91.3±0.5|89.0±0.2|90.7±0.2|91.2±0.4|**91.5±0.6**|
|CheXpert|87.1±0.3|81.9±0.5|86.5±0.3|77.9±0.4|75.7±1.7|82.2±5.1|**90.0±0.2**|
**Table 1b: Avg. acc.**


|Worst group acc.|difFOCI+ERM|difFOCI+DRO|ERM|DRO|JTT|Mixup|IRM|
|--|--|--|--|--|---|---|--|
|MultiNLI|**77.6±0.1**|**77.5±0.2**|66.9±0.5|77.0±0.1|69.6±0.1|69.5±0.4|66.5±1.0|
|CivilComments|66.32±0.2|**70.3±0.2**|64.1±1.1|**70.2±0.8**|64.0±1.1|65.1±0.9|63.2±0.5|
|CelebA|**89.32±0.4**|**89.8±0.9**|65.0±2.5|**88.8±0.6**|70.3±0.5|57.6±0.5|63.1±1.7|
|NICO++|**47.10±0.7**|46.3±0.2|39.3±2.0|38.3±1.2|40.0±0.0|43.1±0.7|40.0±0.0|
|MetaShift|83.10±0.5|**91.7±0.2**|80.9±0.3|86.2±0.6|82.6±0.6|80.9±0.8|84.0±0.4|
|CheXpert|54.42±3.2|**75.3±0.3**|50.1±3.5|73.9±0.4|61.5±4.3|40.2±4.1|35.1±1.2|
**Table 1c: Worst group acc.**

To address the second point, we expanded analysis in Section 5.3 showing that (dF3) maintains good predictive performance of $y$ (from $X$), while also significantly decreasing predictability about sensitive attributes $X_s$.

Specifically, we train two NNs to predict $y$: the first NN was trained on $X$ (without $X_s$), while the second NN was trained on features $f_\theta(X)$ obtained using (dF3). We then used the final layers of both NNs to predict the sensitive $X_s$ and found that difFOCI (dF3) significantly reduced the predictability of $X_s$ (sometimes to chance level) without significantly impacting accuracy on $y$ (in some cases even slightly improving it). The results are presented in Table 2 below:


|Dataset|Features|Train acc: $y$|Val. Acc: $y$|Test acc: $y$|Train acc: $X_s$|Val. Acc: $X_s$|Test acc: $X_s$|
|--|--|--|--|---|--|--|--|
|Bank marketing|Stand. data|91.32±2.3|93.27±1.2|90.05±2.0|89.09±1.2|72.26±1.5|70.93±0.9|
||(dF3) features|90.81±1.8|92.13±2.6|89.35±1.1|63.12±2.8|62.24±0.7|**63.81±2.1**|
|Student data|Stand. data|88.35±1.7|79.63±0.9|75.67±1.3|95.68±2.1|72.16±2.4|71.21±1.5|
||(dF3) features|80.18±2.9|72.16±1.6|72.73±1.7|59.47±1.1|58.95±1.0|**48.89±1.1**|
|ASCI Income|Stand. data|83.49±2.4|85.10±2.1|81.30±2.7|68.97±1.6|67.67±2.6|66.00±0.7|
||(dF3) features|82.80±0.8|81.99±1.5|82.95±0.9|56.58±1.2|55.01±2.0|**52.73±2.0**|
**Table 2. Fairness experiments**




While these additions did not substantially alter the manuscript's content, apart from rephrasing subsection 5.3 in response to Reviewer uzzo's valid remark, we believe they considerably strengthened the paper. We extend our gratitude to all the reviewers for their contributions to these improvements.

---

[1] Liu, E.Z., Haghgoo, B., Chen, A.S., Raghunathan, A., Koh, P.W., Sagawa, S., Liang, P. and Finn, C., 2021, July. Just train twice: Improving group robustness without training group information.

[2] Hongyi Zhang, Moustapha Cisse, Yann N Dauphin, and David Lopez-Paz. Mixup: Beyond empirical risk minimization.

[3] Arjovsky, M., Bottou, L., Gulrajani, I. and Lopez-Paz, D., 2019. Invariant risk minimization.

---

### Author Response · Authors · 2024-11-25

We appreciate the time and effort taken by all reviewers to provide feedback on our paper. We have now responded to all comments and uploaded a revised manuscript. If there are any other questions or concerns, we would be happy to address them.

---

### Meta-Review · Area_Chair_Qwp7 · 2024-12-23

**Metareview:**

Here the authors propose a new algorithm called diffFOCI which is a differentiable approximation to a recently proposed variable selection method called FOCI (Azadkia and Chatterjee (2021)). The authors combine diffFOCI with NN end-to-end training and show promising results in three very important areas -- feature selection, domain-shift/spurious-correlation, and fairness.

All the reviewers unanimously gave a final score of 6. Though they appreciated the underlying idea, motivation, and the writing quality; each raised more or less the same concern which was regarding the limited experimental results. However, during the rebuttal period the authors did a wonderful job in answering most of their queries and presented new set of promising results on a variety of new datasets and artchitectures (BERT, DINO, CLIP etc.). There were some concerns regarding the theorem and the independence assumptions which also the authors replied convincingly.

I think that this paper is a very nice blend of theory and practice and the results are very promising. Of course the proposed method is not an absolute winner in all the tasks presented in this work but still it seems to be the one with the most promising results throughout. Hence, I support the acceptance of this work and would strongly suggest the reviewers to incorporate all the comments by the reviewers in their revised draft. They all makes sense and would surely help in improving the quality/visibility of the work.

**Additional Comments On Reviewer Discussion:**

All the reviewers primarily showed concerns due to the limited nature of the experiments (which I completely agree with), however, during the rebuttal period the authors did a wonderful job in presenting new set of promising results on a variety of datasets (NICO++, MetaShift, etc.) and architectures (BERT, DINO, CLIP etc.). There were additional experiments to show effectiveness in reducing bias and spurious correlations as well. Some concerns regarding the theorem and the independence assumptions was also replied convincingly by the authors.

---

### Decision · Program_Chairs · 2025-01-22

Accept (Poster)